# Spectral Learning of Large Structured HMMs for Comparative Epigenomics

**Chicheng Zhang**
UC San Diego
chz038@eng.ucsd.edu

**Jimin Song**
Rutgers University
song@dls.rutgers.edu

**Kevin C Chen**
Rutgers University
kcchen@dls.rutgers.edu

**Kamalika Chaudhuri**
UC San Diego
kamalika@eng.ucsd.edu

## Abstract

We develop a latent variable model and an efficient spectral algorithm motivated by the recent emergence of very large data sets of chromatin marks from multiple human cell types. A natural model for chromatin data in one cell type is a Hidden Markov Model (HMM); we model the relationship between multiple cell types by connecting their hidden states by a fixed tree of known structure.

The main challenge with learning parameters of such models is that iterative methods such as EM are very slow, while naive spectral methods result in time and space complexity exponential in the number of cell types. We exploit properties of the tree structure of the hidden states to provide spectral algorithms that are more computationally efficient for current biological datasets. We provide sample complexity bounds for our algorithm and evaluate it experimentally on biological data from nine human cell types. Finally, we show that beyond our specific model, some of our algorithmic ideas can be applied to other graphical models.

## 1 Introduction

In this paper, we develop a latent variable model and efficient spectral algorithm motivated by the recent emergence of very large data sets of chromatin marks from multiple human cell types [7, 9]. Chromatin marks are chemical modifications on the genome which are important in many basic biological processes. After standard preprocessing steps, the data consists of a binary vector (one bit for each chromatin mark) for each position in the genome and for each cell type.

A natural model for chromatin data in one cell type is a Hidden Markov Model (HMM) [8, 13], for which efficient spectral algorithms are known. On biological data sets, spectral algorithms have been shown to have several practical advantages over maximum likelihood-based methods, including speed, prediction accuracy and biological interpretability [24]. Here we extend the approach by modeling multiple cell types together. We model the relationships between cell types by connecting their hidden states by a fixed tree, the standard model in biology for relationships between cell types. This comparative approach leverages the information shared between the different data sets in a statistically unified and biologically motivated manner.

Formally, our model is an HMM where the hidden state $z_t$ at time $t$ has a structure represented by a tree graphical model of known structure. For each tree node $u$ we can associate an individual hidden state $z_t^u$ that depends not only on the previous hidden state $z_{t-1}^u$ for the same tree node $u$ but also on the individual hidden state of its parent node. Additionally, there is an observation variable $x_t^u$ for each node $u$, and the observation $x_t^u$ is independent of other state and observation variables

conditioned on the hidden state variable $z_t^u$. In the bioinformatics literature, [5] studied this model with the additional constraint that all tree nodes share the same emission parameters. In biological applications, the main outputs of interest are the learned observation matrices of the HMM and a segmentation of the genome into regions which can be used for further studies.

A standard approach to unsupervised learning of HMMs is the Expectation-Maximization (EM) algorithm. When applied to HMMs with very large state spaces, EM is very slow. A recent line of work on spectral learning [18, 1, 23, 6] has produced much more computationally efficient algorithms for learning many graphical models under certain mild conditions, including HMMs. However, a naive application of these algorithms to HMMs with large state spaces results in computational complexity exponential in the size of the underlying tree.

Here we exploit properties of the tree structure of the hidden states to provide spectral algorithms that are more computationally efficient for current biological datasets. This is achieved by three novel key ideas. Our first key idea is to show that we can treat each root-to-leaf path in the tree separately and learn its parameters using tensor decomposition methods. This step improves the running time because our trees typically have very low depth. Our second key idea is a novel tensor symmetrization technique that we call *Skeletensor construction* where we avoid constructing the full tensor over the entire root-to-leaf path. Instead we use carefully designed symmetrization matrices to reveal its range in a *Skeletensor* which has dimension equal to that of a single tree node. The third and final key idea is called *Product Projections*, where we exploit the independence of the emission matrices along the root-to-leaf path conditioned on the hidden states to avoid constructing the full tensors and instead construct compressed versions of the tensors of dimension equal to the number of hidden states, not the number of observations. Beyond our specific model, we also show that Product Projections can be applied to other graphical models and thus we contribute a general tool for developing efficient spectral algorithms.

Finally we implement our algorithm and evaluate it on biological data from nine human cell types [7]. We compare our results with the results of [5] who used a variational EM approach. We also compare with spectral algorithms for learning HMMs for each cell type individually to assess the value of the tree model.

## 1.1 Related Work

The first efficient spectral algorithm for learning HMM parameters was due to [18]. There has been an explosion of follow-up work on spectral algorithms for learning the parameters and structure of latent variable models [23, 6, 4]. [18] gives a spectral algorithm for learning an *observable operator* representation of an HMM under certain rank conditions. [23] and [3] extend this algorithm to the case when the transition matrix and the observation matrix respectively are rank-deficient. [19] extends [18] to Hidden Semi-Markov Models.

[2] gives a general spectral algorithm for learning parameters of latent variable models that have a *multi-view* structure – there is a hidden node and three or more observable nodes that are not connected to any other nodes and are independent conditioned on the hidden node. Many latent variable models have this structure, including HMMs, tree graphical models, topic models and mixture models. [1] provides a simpler, more robust algorithm that involves decomposing a third order tensor. [21, 22, 25] provide algorithms for learning latent trees and of latent junction trees.

Several algorithms have been designed for learning HMM parameters for chromatin modeling, including stochastic variational inference [16] and contrastive learning of two HMMs [26]. However, none of these methods extend directly to modeling multiple chromatin sequences simultaneously.

## 2 The Model

**Probabilistic Model.** The natural probabilistic model for a single epigenomic sequence is a hidden Markov model (HMM), where time corresponds to position in the sequence. The observation at time $t$ is the sequence value at position $t$, and the hidden state at $t$ is the regulatory function in this position.

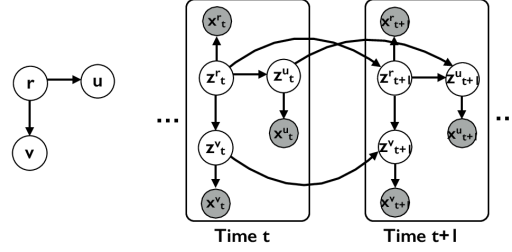

Figure 1: Left: A tree $T$ with 3 nodes $V = \{r, u, v\}$. Right: A HMM whose hidden state has structure $T$.

In comparative epigenomics, the goal is to jointly model epigenomic sequences from multiple species or cell-types. This is done by an HMM with a tree-structured hidden state [5](THS-HMM),[1] where each node in the tree representing the hidden state has a corresponding observation node. Formally, we represent the model by a tuple $\mathcal{H} = (G, \mathcal{O}, \mathcal{T}, \mathcal{W})$; Figure 1 shows a pictorial representation.

$G = (V, E)$ is a directed tree with known structure whose nodes represent individual cell-types or species. The hidden state $z_t$ and the observation $x_t$ are represented by vectors $\{z_t^u\}$ and $\{x_t^u\}$ indexed by nodes $u \in V$. If $(v, u) \in E$, then $v$ is the parent of $u$, denoted by $\pi(u)$; if $v$ is a parent of $u$, then for all $t$, $z_t^v$ is a parent of $z_t^u$. In addition, the observations have the following product structure: if $u' \neq u$, then conditioned on $z_t^u$, the observation $x_t^u$ is independent of $z_t^{u'}$ and $x_t^{u'}$ as well as any $z_{t'}^{u'}$ and $x_{t'}^{u'}$ for $t \neq t'$.

$\mathcal{O}$ is a set of observation matrices $O^u = P(x_t^u | z_t^u)$ for each $u \in V$ and $\mathcal{T}$ is a set of transition tensors $T^u = P(z_{t+1}^u | z_t^u, z_{t+1}^{\pi(u)})$ for each $u \in V$. Finally, $\mathcal{W}$ is the set of initial distributions where $W^u = P(z_1^u | z_1^{\pi(u)})$ for each $z_1^u$.

Given a tree structure and a number of iid observation sequences corresponding to each node of the tree, our goal is to determine the parameters of the underlying THS-HMM and then use these parameters to infer the most likely regulatory function at each position in the sequences.

Below we use the notation $D$ to denote the number of nodes in the tree and $d$ to denote its depth. For typical epigenomic datasets, $D$ is small to moderate (5-50) while $d$ is very small (2 or 3) as it is difficult to obtain data with large $d$ experimentally. Typically $m$, the number of possible values assumed by the hidden state at a single node, is about 6-25, while $n$, the number of possible observation values assumed by a single node is much larger (e.g. 256 in our dataset).

**Tensors.** An order-3 tensor $M \in \mathbb{R}^{n_1} \otimes \mathbb{R}^{n_2} \otimes \mathbb{R}^{n_3}$ is a 3-dimensional array with $n_1 n_2 n_3$ entries, with its $(i_1, i_2, i_3)$-th entry denoted as $M_{i_1, i_2, i_3}$.

Given $n_i \times 1$ vectors $v_i$, $i = 1, 2, 3$, their tensor product, denoted by $v_1 \otimes v_2 \otimes v_3$ is the $n_1 \times n_2 \times n_3$ tensor whose $(i_1, i_2, i_3)$-th entry is $(v_1)_{i_1}(v_2)_{i_2}(v_3)_{i_3}$. A tensor that can be expressed as the tensor product of a set of vectors is called a rank 1 tensor. A tensor $M$ is symmetric if and only if for any permutation $\pi : [3] \to [3]$, $M_{i_1, i_2, i_3} = M_{\pi(i_1), \pi(i_2), \pi(i_3)}$.

Let $M \in \mathbb{R}^{n_1} \otimes \mathbb{R}^{n_2} \otimes \mathbb{R}^{n_3}$. If $V_i \in \mathbb{R}^{n_i \times m_i}$, then $M(V_1, V_2, V_3)$ is a tensor of size $m_1 \times m_2 \times m_3$, whose $(i_1, i_2, i_3)$-th entry is: $M(V_1, V_2, V_3)_{i_1, i_2, i_3} = \sum_{j_1, j_2, j_3} M_{j_1, j_2, j_3}(V_1)_{j_1, i_1}(V_2)_{j_2, i_2}(V_3)_{j_3, i_3}$.

Since a matrix is a order-2 tensor, we also use the following shorthand to denote matrix multiplication. Let $M \in \mathbb{R}^{n_1} \otimes \mathbb{R}^{n_2}$. If $V_i \in \mathbb{R}^{m_i \times n_i}$, then $M(V_1, V_2)$ is a matrix of size $m_1 \times m_2$, whose $(i_1, i_2)$-th entry is: $M(V_1, V_2)_{i_1, i_2} = \sum_{j_1, j_2} M_{j_1, j_2}(V_1)_{j_1, i_1}(V_2)_{j_2, i_2}$. This is equivalent to $V_1^\top M V_2$.

**Meta-States and Observations, Co-occurrence Matrices and Tensors.** Given observations $x_t^u$ and $x_{t'}^u$ at a single node $u$, we use the notation $P_{t,t'}^u$ to denote their expected co-occurence frequencies: $P_{t,t'}^{u,u} = \mathbb{E}[x_t^u \otimes x_{t'}^u]$, and $\hat{P}_{t,t'}^{u,u}$ to denote their corresponding empirical version. The tensor $P_{t,t',t''}^{u,u,u} = \mathbb{E}[x_t^u \otimes x_{t'}^u \otimes x_{t''}^u]$ and its empirical version $\hat{P}_{t,t',t''}^{u,u,u}$ are defined similarly.

Occasionally, we will consider the states or observations corresponding to a subset of nodes in $G$ coalesced into a single meta-state or meta-observation. Given a connected subset $S \subseteq V$ of nodes in the tree $G$ that includes the root, we use the notation $z_t^S$ and $x_t^S$ to denote the meta-state represented by $(z_t^u, u \in S)$ and the meta-observation represented by $(x_t^u, u \in S)$ respectively. We define the observation matrix for $S$ as $O^S = P(x_t^S | z_t^S) \in \mathbb{R}^{n^{|S|} \times m^{|S|}}$ and the transition matrix for $S$ as $T^S = P(z_{t+1}^S | z_t^S) \in \mathbb{R}^{m^{|S|} \times m^{|S|}}$, respectively.

For sets of nodes $V_1$ and $V_2$, we use the notation $P_{t,t'}^{V_1,V_2}$ to denote the expected co-occurrence frequencies of the meta-observations $x_t^{V_1}$ and $x_{t'}^{V_2}$. Its empirical version is denoted by $\hat{P}_{t,t'}^{V_1,V_2}$. Similarly, we can define the notation $P_{t,t',t''}^{V_1,V_2,V_3}$ and its empirical version $\hat{P}_{t,t',t''}^{V_1,V_2,V_3}$.

**Background on Spectral Learning for Latent Variable Models.** Recent work by [1] has provided a novel elegant tensor decomposition method for learning latent variable models. Applied to HMMs, the main idea is to decompose a transformed version of the third order co-occurrence tensor of the first three observations to recover the parameters; [1] shows that given enough samples and under fairly mild conditions on the model, this provides an approximation to the globally optimal solution. The algorithm has three main steps. First, the third order tensor of the co-occurrences is symmetrized using the second order co-occurrence matrices to yield a symmetric tensor; this symmetric tensor is then orthogonalized by a whitening transformation. Finally, the resultant symmetric orthogonal tensor is decomposed via the tensor power method.

In biological applications, instead of multiple independent sequences, we have a single long sequence in the steady state. In this case, following ideas from [23], we use the average over $t$ of the third order co-occurence tensors of three consecutive observations starting at time $t$. The second order co-occurence tensor is also modified similarly.

# 3   Algorithm

A naive approach for learning parameters of HMMs with tree-structured hidden states is to directly apply the spectral method of [1]. Since this method ignores the structure of the hidden state, its running time is very high, $\Omega(n^D m^D)$, even with optimized implementations. This motivates the design of more computationally efficient approaches.

A plausible approach is to observe that at $t = 1$, the observations are generated by a tree graphical model; thus in principle one could learn the parameters of the underlying tree using existing algorithms [22, 21, 25]. However, this approach does not directly produce the HMM parameters; it also does not work for biological sequences because we do not have multiple independent samples at $t = 1$; instead we have a single long sequence at the steady state, and the steady state distribution of observations is not generated by a latent tree. Another plausible approach is to use the spectral junction tree algorithm of [25]; however, this algorithm does not provide the actual transition and observation matrix parameters which hold important biological information, and instead provides an *observable operator representation*.

Our main contribution is to show that we can achieve a much better running time by exploiting the structure of the hidden state. Our algorithm is based on three key ideas – Partitioning, Skeletensor Construction and Product Projections. We explain these ideas next.

**Partitioning.** Our first observation is that to learn the parameters at a node $u$, we can focus only on the unique path from the root to $u$. Thus we *partition* the learning problem on the tree into separate learning problems on these paths. This maintains correctness as proved in the Appendix.

The Partitioning step reduces the computational complexity since we now need to learn an HMM with $m^d$ states and $n^d$ observations, instead of the naive method where we learn an HMM with $m^D$ states and $n^D$ observations. As $d \ll D$ in biological data, this gives us significant savings.

**Constructing the Skeletensor.** A naive way to learn the parameters of the HMM corresponding to each root-to-node path is to work directly on the $O(n^d \times n^d \times n^d)$ co-occurrence tensor. Instead, we show that for each node $u$ on a root-to-node path, a novel symmetrization method can be used to construct a much smaller *skeleton tensor* $T^u$ of size $n \times n \times n$, which nevertheless captures the effect of the entire root-to-node path and projects it into the skeleton tensor, thus revealing the range of $O^u$. We call this the *skeletensor*.

Let $H_u$ be the path from the root to a node $u$, and let $\hat{P}_{1,2,3}^{H_u,u,H_u}$ be the empirical $n^{|H_u|} \times n \times n^{|H_u|}$ tensor of co-occurrences of the meta-observations $H_u$, $u$ and $H_u$ at times 1, 2 and 3 respectively. Based on the data we construct the following symmetrization matrices:

$$S_1 \sim \hat{P}_{2,3}^{u,H_u}(\hat{P}_{1,3}^{H_u,H_u})^\dagger, \quad S_3 \sim \hat{P}_{2,1}^{u,H_u}(\hat{P}_{3,1}^{H_u,H_u})^\dagger$$

Note that $S_1$ and $S_3$ are $n \times n^{|H_u|}$ matrices. Symmetrizing $\hat{P}_{1,2,3}^{H_u,u,H_u}$ with $S_1$ and $S_3$ gives us an $n \times n \times n$ skeletensor, which can in turn be decomposed to give an estimate of $O^u$ (see Lemma 3 in the Appendix).

Even though naively constructing the symmetrization matrices and skeletensor takes $O(Nn^{2d+1} + n^{3d})$ time, this procedure improves computational efficiency because tensor construction is a one-time operation, while the power method which takes many iterations is carried out on a much smaller tensor.

**Product Projections.** We further reduce the computational complexity by using a novel algorithmic technique that we call *Product Projections*. The key observation is as follows. Let $H_u = \{u_0, u_1, \dots, u_{d-1}\}$ be any root-to-node path in the tree and consider the HMM that generates the observations $(x_t^{u_0}, x_t^{u_1}, \dots, x_t^{u_{d-1}})$ for $t = 1, 2, \dots$. Even though the individual observations $x_t^{u_j}, j = 0, 1, \dots, d-1$ are highly dependent, *the range of $O^{H_u}$, the emission matrix of the HMM describing the path $H_u$, is contained in the product of the ranges of $O^{u_j}$*, where $O^{u_j}$ is the emission matrix at node $u_j$ (Lemma 4 in the Appendix). Furthermore, even though the $O^{u_j}$ matrices are difficult to find, *their ranges can be determined by computing the SVDs of the observation co-occurrence matrices at $u_j$*.

Thus we can *implicitly* construct and store (an estimate of) the range of $O^{H_u}$. This also gives us estimates of the range of $\hat{P}_{1,3}^{H_u,H_u}$, the column spaces of $\hat{P}_{2,1}^{u,H_u}$ and $\hat{P}_{2,3}^{u,H_u}$, and the range of the first and third modes of the tensor $\hat{P}_{1,2,3}^{H_u,u,H_u}$. Therefore during skeletensor construction we can avoid explicitly constructing $S_1$, $S_3$ and $\hat{P}_{1,2,3}^{H_u,u,H_u}$, and instead construct their projections onto their ranges. This reduces the time complexity of the skeletensor construction step to $O(Nm^{2d+1} + m^{3d} + dmn^2)$ (recall that the range has dimension $m$.) While the number of hidden states $m$ could be as high as $n$, this is a significant gain in practice, as $n \gg m$ in biological datasets (e.g. 256 observations vs. 6 hidden states).

Product projections are more efficient than random projections [17] on the co-occurrence matrix of meta-observations: the co-occurrence matrices are $n^d \times n^d$ matrices, and random projections would take $\Omega(n^d)$ time. Also, product projections differ from the suggestion of [15] since we exploit properties of the model to efficiently find good projections.

The Product Projections technique is a general technique with applications beyond our model. Some examples are provided in Appendix C.3.

## 3.1 The Full Algorithm

Our final algorithm follows from combining the three key ideas above. Algorithm 1 shows how to recover the observation matrices $O^u$ at each node $u$. Once the $O^u$s are recovered, one can use standard techniques to recover $T$ and $W$; details are described in Algorithm 2 in the Appendix.

## 3.2 Performance Guarantees

We now provide performance guarantees on our algorithm. Since learning parameters of HMMs and many other graphical models is NP-Hard, spectral algorithms make simplifying assumptions on the properties of the model generating the data. Typically these assumptions take the form of some

**Algorithm 1** Algorithm for Observation Matrix Recovery

---

1: Input: $N$ samples of the three consecutive observations $(x_1, x_2, x_3)_{i=1}^N$ generated by an HMM with tree structured hidden state with known tree structure.
2: **for** $u \in V$ **do**
3:    Perform SVD on $\hat{P}_{1,2}^{u,u}$ to get the first $m$ left singular vectors $\hat{U}^u$.
4: **end for**
5: **for** $u \in V$ **do**
6:    Let $H_u$ denote the set of nodes on the unique path from root $r$ to $u$. Let $\hat{U}^{H_u} = \otimes_{v \in H_u} \hat{U}^v$.
7:    **Construct Projected Skeletensor.** First, compute symmetrization matrices:

$$\hat{S}_1^u = ((\hat{U}^u)^\top \hat{P}_{2,3}^{u,H_u} \hat{U}^{H_u})((\hat{U}^{H_u})^\top \hat{P}_{1,3}^{H_u,H_u} \hat{U}^{H_u})^{-1}$$

$$\hat{S}_3^u = ((\hat{U}^u)^\top \hat{P}_{2,1}^{u,H_u} \hat{U}^{H_u})((\hat{U}^{H_u})^\top \hat{P}_{3,1}^{H_u,H_u} \hat{U}^{H_u})^{-1}$$

8:    Compute symmetrized second and third co-occurrences for $u$:

$$\hat{M}_2^u = (\hat{P}_{1,2}^{H_u,u}(\hat{U}^{H_u}(\hat{S}_1^u)^\top, \hat{U}^u) + \hat{P}_{1,2}^{H_u,u}(\hat{U}^{H_u}(\hat{S}_1^u)^\top, \hat{U}^u)^\top)/2$$

$$\hat{M}_3^u = \hat{P}_{1,2,3}^{H_u,u,H_u}(\hat{U}^{H_u}(\hat{S}_1^u)^\top, \hat{U}^u, \hat{U}^{H_u}(\hat{S}_3^u)^\top)$$

9:    **Orthogonalization and Tensor Decomposition.** Orthogonalize $\hat{M}_3^u$ using $\hat{M}_2^u$ and decompose to recover $(\hat{\theta}_1^u, \ldots, \hat{\theta}_m^u)$ as in [1] (See Algorithm 3 in the Appendix for details).
10:    **Undo Projection onto Range.** Estimate $O^u$ as: $\hat{O}^u = \hat{U}^u \hat{\Theta}^u$, where $\hat{\Theta}^u = (\hat{\theta}_1^u, \ldots, \hat{\theta}_m^u)$.
11: **end for**

---

conditions on the rank of certain parameter matrices. We state below the conditions needed for our algorithm to successfully learn parameters of a HMM with tree structured hidden states. Observe that we need two kinds of rank conditions – node-wise and path-wise – to ensure that we can recover the full set of parameters on a root-to-node path.

**Assumption 1** (Node-wise Rank Condition). *For all $u \in V$, the matrix $O^u$ has rank $m$, and the joint probability matrix $P_{2,1}^{u,u}$ has rank $m$.*

**Assumption 2** (Path-wise Rank Condition). *For any $u \in V$, let $H_u$ denote the path from root to $u$. Then, the joint probability matrix $P_{1,2}^{H_u,H_u}$ has rank $m^{|H_u|}$.*

Assumption 1 is required to ensure that the skeletensor can be decomposed, and that $\hat{U}^u$ indeed captures the range of $O^u$. Assumption 2 ensures that the symmetrization operation succeeds. This kind of assumption is very standard in spectral learning [18, 1].

[3] has provided a spectral algorithm for learning HMMs involving fourth and higher order moments when Assumption 1 does not hold. We believe similar approaches will apply to our problem as well, and we leave this as an avenue for future work.

If Assumptions 1 and 2 hold, we can show that Algorithm 1 is consistent – provided enough samples are available, the model parameters learnt by the algorithms are close to the true model parameters. A finite sample guarantee is provided in the Appendix.

**Theorem 1** (Consistency). *Suppose we run Algorithm 1 on the first three observation vectors $\{x_{i,1}, x_{i,2}, x_{i,3}\}$ from $N$ iid sequences generated by an HMM with tree-structured hidden states. Then, for all nodes $u \in V$, the recovered estimates $\hat{O}^u$ satisfy the following property: with high probability over the iid samples, there exists a permutation $\Pi^u$ of the columns of $\hat{O}^u$ such that as $\|O^u - \Pi^u \hat{O}^u\| \leq \varepsilon(N)$ where $\varepsilon(N) \to 0$ as $N \to \infty$.*

Observe that the observation matrices (as well as the transition and initial probabilities) are recovered upto permutations of hidden states *in a globally consistent manner*.

# 4 Experiments

**Data and experimental settings.** We ran our algorithm, which we call "Spectacle-Tree", on a chromatin dataset on human chromosome 1 from nine cell types (H1-hESC, GM12878, HepG2, HMEC, HSMM, HUVEC, K562, NHEK, NHLF) from the ENCODE project [7]. Following [5], we used a biologically motivated tree structure of a star tree with H1-hESC, the embryonic stem cell type, as the root. There are data for eight chromatin marks for each cell type which we preprocessed into binary vectors using a standard Poisson background assumption [11]. The chromosome is divided into 1,246,253 segments of length 200, following [11]. The observed data consists of a binary vector of length eight for each segment, so the number of possible observations is the number of all combinations of presence or absence of the chromatin marks (i.e. $n = 2^8 = 256$). We set the number of hidden states, which we interpret as chromatin states, to $m = 6$, similar to the choice of ENCODE. Our goals are to discover chromatin states corresponding to biologically important functional elements such as promoters and enhancers, and to label each chromosome segment with the most probable chromatin state.

Observe that instead of the first few observations from $N$ iid sequences, we have a single long sequence in the steady state per cell type; thus, similar to [23], we calculate the empirical co-occurrence matrices and tensors used in the algorithm based on two and three successive observations respectively (so, more formally, instead of $\hat{P}_{1,2}$, we use the average over $t$ of $\hat{P}_{t,t+1}$ and so on). Additionally, we use a projection procedure similar to [4] for rounding negative entries in the recovered observation matrices. Our experiments reveal that the rank conditions appear to be satisfied for our dataset.

**Run time and memory usage comparisons.** First, we flattened the HMM with tree-structured hidden states into an ordinary HMM with an exponentially larger state space. Our Python implementation of the spectral algorithm for HMMs of [18] ran out of memory while performing singular value decomposition on the co-occurence matrix, even using sparse matrix libraries. This suggests that naive application of spectral HMM is not practical for biological data.

Next we compared the performance of Spectacle-Tree to a similar model which additionally constrained all transition and observation parameters to be the same on each branch [5]. That work used several variational approximations to the EM algorithm and reported that SMF (structured mean field) performed the best in their tests. Although we implemented Spectacle-Tree in Matlab and did not optimize it for run-time efficiency, Spectacle-Tree took ∼2 hr, whereas the SMF algorithm took ∼13 hr for 13 iterations to convergence. This suggests that spectral algorithms may be much faster than variational EM for our model.

**Biological interpretation of the observation matrices.** Having examined the efficiency of Spectacle-Tree, we next studied the accuracy of the learned parameters. We focused on the observation matrices which hold most of the interesting biological information. Since the full observation matrix is very large ($2^8 \times 6$ where each row is a combination of chromatin marks), Figure 2 shows the $8 \times 6$ marginal distribution of each chromatin mark conditioned on each hidden state. Spectacle-Tree identified most of the major types of functional elements typically discovered from chromatin data: repressive, strong enhancer, weak enhancer, promoter, transcribed region and background state (states 1-6, respectively, in Figure 2b). In contrast, the SMF algorithm used three out of the six states to model the large background state (i.e. the state with no chromatin marks). It identified repressive, transcribed and promoter states (states 2, 4, 5, respectively, in Figure 2a) but did not identify any enhancer states, which are one of the most interesting classes for further biological studies.

We believe these results are due to that fact that the background state in the data set is large: ∼62% of the segments do not have chromatin marks for any cell type. The background state has lower biological interest but is modeled well by the maximum likelihood approach. In contrast, biologically interesting states such as promoters and enhancers comprise a relatively small fraction of the genome. We cannot simply remove background segments to make the classes balanced because it would change the length distribution of the hidden states. Finally, we observed that our model estimated significantly different parameters for each cell type which captures different chromatin states (Appendix Figure 3). For example, we found enhancer states with strong H3K27ac in all cell types except for H1-hESC, where both enhancer states (3 and 6) had low signal for this mark. This mark is known to be biologically important in these cells for distinguishing active from poised enhancers

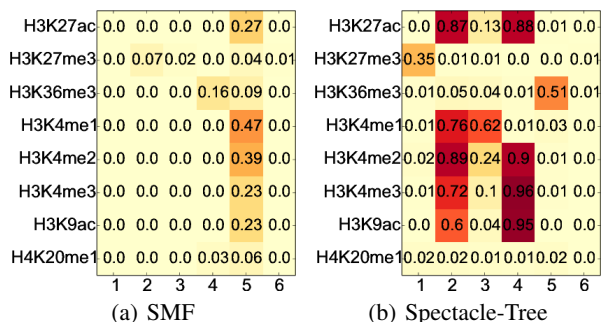

|         | 1 | 2 | 3 | 4 | 5 | 6 |
|---------|---|---|---|---|---|---|
(a) SMF          (b) Spectacle-Tree

Figure 2: The compressed observation matrices for the GM12878 cell type estimated by the SMF and Spectacle-Tree algorithms. The hidden states are on the X axis.

[10]. This suggests that modeling the additional branch-specific parameters can yield interesting biological insights.

**Comparison of the chromosome segments labels.** We computed the most probable state for each chromosome segment using a posterior decoding algorithm. We tested the accuracy of the predictions using an experimentally defined data set and compared it to SMF and the spectral algorithm for HMMs run for individual cell types without the tree (Spectral-HMM). Specifically we assessed promoter prediction accuracy (state 5 for SMF and state 4 for Spectacle-Tree in Figure 2) using CAGE data from [14] which was available for six of the nine cell types. We used the F1 score (harmonic mean of precision and recall) for comparison and found that Spectacle-Tree was much more accurate than SMF for all six cell types (Table 1). This was because the promoter predictions of SMF were biased towards the background state so those predictions had slightly higher recall but much lower specificity.

Finally, we compared our predictions to Spectral-HMM to assess the value of the tree model. H1-hESC is the root node so Spectral-HMM and Spectacle-Tree have the same model and obtain the same accuracy (Table 1). Spectacle-Tree predicts promoters more accurately than Spectral-HMM for all other cell types except HepG2. However, HepG2 is the most diverged from the root among the cell types based on the Hamming distance between the chromatin marks. We hypothesize that for HepG2, the tree is not a good model which slightly reduces the prediction accuracy.

| Cell type | SMF | Spectral-HMM | Spectacle-Tree |
|-----------|-----|--------------|----------------|
| H1-hESC | .0273 | **.1930** | **.1930** |
| GM12878 | .0220 | .1230 | **.1703** |
| HepG2 | .0274 | **.1022** | .0993 |
| HUVEC | .0275 | .1221 | **.1621** |
| K562 | .0255 | .0964 | **.1966** |
| NHEK | .0287 | .1528 | **.1719** |

Table 1: F1 score for predicting promoters for six cell types. The highest F1 score for each cell type is emphasized in bold. Ground-truth labels for the other 3 cell-types are currently unavailable.

Our experiments show that Spectacle-Tree has improved computational efficiency, biological interpretability and prediction accuracy on an experimentally-defined feature compared to variational EM for a similar tree HMM model and a spectral algorithm for single HMMs. A previous study showed improvements for spectral learning of single HMMs over the EM algorithm [24]. Thus our algorithms may be useful to the bioinformatics community in analyzing the large-scale chromatin data sets currently being produced.

**Acknowledgements.** KC and CZ thank NSF under IIS 1162581 for research support.

## Footnotes

[1] In the bioinformatics literature, this model is also known as a tree HMM.

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
