[Supplementary Material]

# A Recovering the Transition Probabilities and Initial Probabilities

Algorithm 2 recovers the transition and initial probabilities, given estimates of observation matrices. Theorem 2 provides finite sample guarantees on Algorithm 1 in conjunction with Algorithm 2.

---

**Algorithm 2** Recovering the Transition Probabilities and Initial Probabilities

---

1: Input: $N$ samples of the first three observations $(x_1, x_2, x_3)_{i=1}^N$ generated by a tree HMM, Estimates of observation matrices $\hat{O}^u$.
2: **for** $u \in V$ **do**
3:    **if** $u$ is root $r$ **then**
4:       Compute $\hat{W}^r = (\hat{O}^u)^\dagger \hat{P}_1^r$.
5:       Compute $\hat{Q}^r = (\hat{O}^r)^\dagger \hat{P}_{2,1}^{r,r} (\hat{O}^r)^{\dagger\top}$.
6:       Normalize over the $z_2^u$ coordinate to get $\hat{T}^u$.
7:    **else**
8:       Compute $\hat{W}^u = (\hat{O}^u)^\dagger P_{1,1}^{u,\pi(u)} (\hat{O}^{\pi(u)})^{\dagger\top}$.
9:       Compute $\hat{Q}^u = P_{2,2,1}^{u,\pi(u),u}((\hat{O}^u)^{\dagger T}, (\hat{O}^{\pi(u)})^{\dagger\top}, (\hat{O}^u)^{\dagger\top})$.
10:       Normalize over the $z_2^u$ coordinate to get $\hat{T}^u$.
11:    **end if**
12: **end for**

---

# B Additional Notations

For a node $u \in V$, when it is clear from context, we sometimes use $H$ to denote $H_u$ and $d$ to denote $d_u$.

Define $O_2^H$ to be a $n^d \times m^d$ matrix whose rows are indexed by elements in $[n]^d$ and columns are indexed by elements in $[m]^d$. In particular, $(O_2^H)_{(i_1,\ldots,i_d),(j_1,\ldots,j_d)} = P(x_2^r = i_1, \ldots, x_2^u = i_d | z_2^r = j_1, \ldots, z_2^u = j_d)$. Similarly we define $O_3^H$ whose entries are $(O_3^H)_{(i_1,\ldots,i_d),(j_1,\ldots,j_d)} = P(x_3^r = i_1, \ldots, x_3^u = i_d | z_2^r = j_1, \ldots, z_2^u = j_d)$, and $O_1^H$ whose its entries are $(O_1^H)_{(i_1,\ldots,i_d),(j_1,\ldots,j_d)} = P(x_1^r = i_1, \ldots, x_1^u = i_d | z_2^r = j_1, \ldots, z_2^u = j_d)$. We define $O_2^u$ to be a $n \times m^d$ matrix, whose rows are indexed by elements in $[n]$, and columns are indexed by elements in $[m]^d$. Its entries are $(O_2^u)_{i,(j_1,\ldots,j_d)} = P(x_1^u = i | z_2^r = j_1, \ldots, z_2^u = j_d)$.

Define $\pi^H$ to be a vector representing the marginal probability of $(z_2^r, \ldots, z_2^u)$. In particular, its rows are indexed by elements in $[m]^d$, and $\pi^H_{(i_1,\ldots,i_d)} = P(z_2^r = i_1, \ldots, z_2^u = i_d)$. Define $\pi^u$ to be a vector representing the marginal probability of $z_2^u$. In particular, its rows are indexed by elements in $[m]$, and $\pi_i^u = P(z_2^u = i)$. Define $\pi_{\min}^u$ as $\min_i \pi_i^u$. Define $\rho^H$ as the $m^d$ dimensional vector representing the marginal probability of $(z_1^r, \ldots, z_1^u)$ whose entries are indexed by elements in $[m]^d$. In particular, $\rho^H_{(i_1,\ldots,i_d)} = P(z_1^r = i_1, \ldots, z_1^u = i_d)$. $T^H$ is defined as the $m^d \times m^d$ matrix representing the conditional probability of $z_2^H$ given $z_1^H$, and its rows and columns are indexed by elements in $[m]^d$, in particular, $T_{(i_1,\ldots,i_d),(j_1,\ldots,j_d)} = P(z_2^r = i_1, \ldots, z_2^u = i_d | z_1^r = j_1, \ldots, z_1^u = j_d)$.

Let $u$ be a node in $V$. Define $U^u$ to be a matrix whose columns form an orthonormal basis of $O^u$. One way to get $U^u$ is to take its columns to be the top $m$ singular vectors of $O^u$. The specific choice of $U^u$ does not affect our analysis, as we will be only looking at the projection matrix $U^u(U^u)^\top$ throughout. Define $U^H$ to be $\otimes_{v \in H} U^u$.

For a matrix $M$, define $\|M\|$ to be its operator norm, that is, $\max_{\|u\|=1,\|v\|=1} \|v^\top M u\|$. Define the Frobenius norm of $M$, $\|M\|_F$ to be square root of the sum of the square of its entries, that is, $\sqrt{\sum_{i,j} M_{ij}^2}$. By standard results in linear algebra, $\|M\| \le \|M\|_F$. Similarly, for a third order tensor $T$, define $\|T\|$ to be its operator norm, that is $\max_{\|u\|=1,\|v\|=1,\|w\|=1} T(u,v,w)$. Define the Frobenius norm of $T$, $\|T\|_F$ to be square root of the sum of the square of its entries, that is, $\sqrt{\sum_{i,j,k} T_{ijk}^2}$. By standard results of linear algebra, $\|T\| \le \|T\|_F$.

## C   Main Lemmas

### C.1   Partitioning Lemmas

**Lemma 1** (Path Partitioning). *Suppose observations and states $\{x_t^v, z_t^v\}_{v \in V, t \in \mathbb{N}}$ are drawn from a THS-HMM represented by $\mathcal{H} = (G, T, O, W)$, where $G = (V, E)$, $T = \{T_v, v \in V\}$, $O = \{O_v, v \in V\}$, $W = \{W_v, v \in V\}$. Let $u \in V$, and let $H_u$ denote nodes inside the unique path from root $r$ to $u$. Then $\{x_t^v, z_t^v\}_{u \in H_u, t \in \mathbb{N}}$ are generated by a THS-HMM represented by a tuple $\tilde{\mathcal{H}} = (\tilde{G}, \tilde{T}, \tilde{O}, \tilde{W})$, where $\tilde{G} = (\tilde{V}, \tilde{E})$ is the induced subgraph on $H_u$. In particular, $\tilde{V} = H_u$, $\tilde{E} = \{(v, \pi(v))\}_{v \in H_u}$, $\tilde{T} = \{T_v, v \in H_u\}$, $\tilde{O} = \{O_v, v \in H_u\}$, $\tilde{W} = \{W_v, v \in H_u\}$.*

*Proof of Lemma 1.* To show this lemma, we will calculate the marginal distribution of the variables $\{x_t^v, z_t^v\}_{v \in H_u, t \in [\tau]}$. Observe that the full joint distribution of $\{x_t^v, z_t^v\}_{v \in G, t \in [\tau]}$ is equal to:

$$\prod_{v \in G} \Pr(z_1^v) \prod_{t=1}^{\tau-1} \prod_{v \in H_u} \Pr(z_{t+1}^v | z_t^v, z_{t+1}^{\pi(v)}) \prod_{t=1}^{\tau} \prod_{v \in G} \Pr(x_t^v | z_t^v)$$

To calculate the marginal over $\{x_t^v, z_t^v\}_{v \in H_u, t \in [\tau]}$, we eliminate the rest of the variables one by one. Observe that we can eliminate any observation variable $x_t^v$ for $v \notin H_u$ without introducing any extra edges, as $x_t^v$ is only connected to $z_t^v$. Moreover, marginalizing $x_t^v$ gives: $\sum_x \Pr(x_t^v = x | z_t^v = z) = 1$.

Let $\tilde{G}$ be the current tree; initially $\tilde{G} = G$. We next eliminate the nodes $\{z_t^v, t = \tau, \ldots, 1\}$ for $v \notin H_u$ one by one where $v \notin H_u$ is a leaf node in $\tilde{G}$. We do this in the order $z_T^v, z_{T-1}^v, \ldots, z_1^v$; once we have eliminated these nodes, we delete $v$ from $\tilde{G}$, and we continue until only the nodes in $H_u$ are left. To eliminate a $z_t^v$ when $\{z_s^v, s > t\}$ have been eliminated, we sum over: $\sum_z \Pr(z_t^v = z | z_{t-1}^v, z_t^{\pi(v)})$ which also sums to 1.

We repeat this process until only the nodes $\{x_t^v, z_t^v\}_{u \in H_u, t \in [T]}$ are left. Since we get 1 from eliminating each variable, the marginal we are left with is:

$$\prod_{v \in H_u} \Pr(z_1^v) \prod_{t=1}^{T-1} \prod_{v \in H_u} \Pr(z_{t+1}^v | z_t^v, z_{t+1}^{\pi(v)}) \prod_{t=1}^{T} \prod_{v \in H_u} \Pr(x_t^v | z_t^v), \tag{1}$$

which is the marginal distribution of an HMM with tree-structured hidden states described by the tuple $(\tilde{G}, \tilde{T}, \tilde{O}, \tilde{W})$. The lemma follows. $\square$

The following is a Corollary of Lemma 1.

**Corollary 1.** *If observations and states $\{x_t^v, z_t^v\}_{v \in H_u, t \in \mathbb{N}}$ are drawn from a THS-HMM represented by $(\tilde{G}, \tilde{T}, \tilde{O}, \tilde{W})$, then the sequence of coalesced observations and states $\{x_t^{H_u}, z_t^{H_u}\}_{t \in \mathbb{N}}$ are drawn from an HMM.*

*Proof.* The proof is a simple extension of Lemma 1. (1) gives us the marginal distribution of $\{x_t^v, z_t^v\}_{v \in H_u, t \in \mathbb{N}}$. Observe that for any $t$, conditioned on $z_t^{H_u}$, $x_t^{H_u}$ is d-separated from all the other nodes of the graph – this is because for any node $x$ in the graphical model, $x_t^{H_u}$, $z_t^{H_u}$ and $x$ either form a chain or or a fork structure whose middle node is $z_t^{H_u}$. Moreover, conditioned on $z_t^{H_u}$, $z_{t+1}^{H_u}$ is d-separated from the set of nodes $\{z_s^{H_u}\}_{s=1}^{t-1}$. This is because $z_s^{H_u}$, $z_t^{H_u}$ and $z_{t+1}^{H_u}$ form a chain structure whose middle node is $z_t^{H_u}$. The lemma thus follows. $\square$

### C.2   Skeletensor Lemmas

In this subsection, we justify our construction of a skeletensor. Let $u$ be any node in the tree $G$ and let $H$ be the path from the root of $G$ to $u$.

Recall that we define $O_1^H$ to be the $n^d \times m^d$ matrix, whose entries are $(O_1^H)_{(i_1,...,i_d),(j_1,...,j_d)} = P(x_1^r = i_1, \ldots, x_1^u = i_d | z_2^r = j_1, \ldots, z_2^u = j_d)$. Similarly, $O_3^H$ is a $n^d \times m^d$ matrix, with entries $(O_3^H)_{(i_1,...,i_d),(j_1,...,j_d)} = P(x_3^r = i_1, \ldots, x_3^u = i_d | z_2^r = j_1, \ldots, z_2^u = j_d)$.

We begin by showing that under Assumptions 1 and 2, the matrices $O_1^H$ and $O_3^H$ for the three-view mixture model induced by the HMM have full column rank.

**Lemma 2.** *Let $u$ be a node in $V$. Recall that $H = H_u$ is the set of nodes along the path from root $r$ to $u$. Then:*
*(1) The matrices $\text{diag}(\rho^H)(T^H)^\top \text{diag}(\pi^H)^{-1}$ and $T^H$ are of full rank.*
*(2) The matrices $O_1^H$ and $O_3^H$ are of full column rank.*

*Proof.* By Lemma 1, $x_1^H$, $x_2^H$, $x_3^H$ are conditionally independent given $h_2^H$. Thus,

$$P_{1,2}^{H,H} = O_1^H \text{diag}(\pi^H)(O_2^H)^\top$$

Since by Assumption 2, $P_{1,2}^{H,H}$ is of rank $m^d$, this implies that the matrix $O_1^H$ must be of rank $m^d$ as well. By Proposition 4.2 of [2],

$$O_1^H = O^H \text{diag}(\rho^H)(T^H)^\top \text{diag}(\pi^H)^{-1}$$

This implies that $\text{diag}(\rho^H)(T^H)^\top \text{diag}(\pi^H)^{-1}$ is of rank $m^d$, which is of full rank. Hence $T^H$ is of full rank. By Proposition 4.2 of [2],

$$O_3^H = O^H T^H$$

This shows $O_3^H$ is of full column rank. $\square$

Second, we discuss the infinite sample version of our symmetrization matrix. This will be extended in Lemma 8 in our detailed finite sample analysis.

**Lemma 3.** *Let $u$ be a node in $V$. Recall that $H_u$ is the set of nodes along the path from root $r$ to $u$. Assume $P_{2,3}^{u,H}, P_{1,3}^{H,H}, P_{2,1}^{u,H}$ are given (where $P_{3,1}^{H,H} = (P_{1,3}^{H,H})^T$). Let the symmetrization matrices be:*

$$S_1^u = P_{2,3}^{u,H}(P_{1,3}^{H,H})^\dagger$$

$$S_3^u = P_{2,1}^{u,H}(P_{3,1}^{H,H})^\dagger$$

*and the ground truth symmetrized pair-wise and triple-wise co-occurence tensors be:*

$$M_2^u = P_{1,2}^{H,u}(S_1^{uT}, I)$$

$$M_3^u = P_{1,2,3}^{H,u,H}(S_1^{uT}, I, S_3^{uT})$$

*Then,*

$$M_2^u = \sum_i \pi_i^u (O^u)_i \otimes (O^u)_i$$

$$M_3^u = \sum_i \pi_i^u (O^u)_i \otimes (O^u)_i \otimes (O^u)_i$$

*Proof.* By Lemma 1, $x_1^H$, $x_2^u$, $x_3^H$ are conditionally independent given $z_2^H$, thus

$$P_{2,3}^{u,H} = O_2^u \text{diag}(\pi^H) O_3^{HT}$$

$$P_{1,3}^{H,H} = O_1^H \text{diag}(\pi^H) O_3^{HT}$$

Lemma 2 implies that $O_1^H$ is of full column rank, and $\text{diag}(\pi^H) O_3^{HT}$ is of full row rank. Therefore by standard properties of pseudoinverse,

$$(P_{1,3}^{H,H})^\dagger = (\text{diag}(\pi^H) O_3^{HT})^\dagger (O_1^H)^\dagger$$

Therefore,

$$S_1^u = O_2^u (O_1^H)^\dagger$$

Likewise,

$$S_3^u = O_2^u(O_3^H)^\dagger$$

Then,

$$
\begin{aligned}
M_2^u &= P_{1,2}^{H,u}(S_1^{uT}, I) \\
&= \sum_{i_1,\ldots,i_D} \pi_{i_1,\ldots,i_D}^H (O_2^u)_{i_1,\ldots,i_D} \otimes (O_2^u)_{i_1,\ldots,i_D} \\
&= \sum_{i_1,\ldots,i_D} \pi_{i_1,\ldots,i_D}^H (O^u)_{i_D} \otimes (O^u)_{i_D} \\
&= \sum_i \pi_i^u (O^u)_i \otimes (O^u)_i
\end{aligned}
$$

$$
\begin{aligned}
M_3^u &= P_{1,2,3}^{H,u,H}(S_1^{uT}, I, S_3^{uT}) \\
&= \sum_{i_1,\ldots,i_D} \pi_{i_1,\ldots,i_D}^H (O_2^u)_{i_1,\ldots,i_D} \otimes (O_2^u)_{i_1,\ldots,i_D} \otimes (O_2^u)_{i_1,\ldots,i_D} \\
&= \sum_{i_1,\ldots,i_D} \pi_{i_1,\ldots,i_D}^H (O^u)_{i_D} \otimes (O^u)_{i_D} \otimes (O^u)_{i_D} \\
&= \sum_i \pi_i^u (O^u)_i \otimes (O^u)_i \otimes (O^u)_i
\end{aligned}
$$

$\square$

## C.3 Product Projections Lemmas

### C.3.1 Application 1: HMM with more general hidden states

Consider an HMM with a hidden state represented by a general graphical model $G = (V, E)$ with an observation variable $x_t^u$ corresponding to each $u \in V$. $x_t^u$ is independent of all other hidden state and observation nodes, conditioned on its corresponding hidden state variable $z_t^u$. In this case, $O^{|V|} = \otimes_{u \in V} O^u$. Similar graphical models have been used in biology to model gene expression time courses [12].

**Lemma 4.** $O^H$, *the observation matrix of the HMM that generates the meta-states and meta-observations* $\{z_t^H, x_t^H\}_{t \in \mathbb{N}}$, *equals* $\bigotimes_{v \in H} O^v$.

*Proof.* We consider the observation matrix of the HMM that generates the meta-states and meta-observations $\{z_t^H, x_t^H\}_{t \in \mathbb{N}}$. The number of possible meta-hidden states $z_t^H$ is $m^d$, indexed by $(z_t^v)_{v \in H}$ and the number of possible meta-observations $x_t^H$ is $n^d$, indexed by $(x_t^v)_{v \in H}$. Thus, the observation matrix $O^H$ is of dimension $n^d \times m^d$. Entrywise,

$$
\begin{aligned}
& (O^{H_u})_{(i_1,\ldots,i_d),(j_1,\ldots,j_d)} \\
&= \mathbb{P}(x_t^r = i_1, \ldots, x_t^u = i_d | z_t^r = j_1, \ldots, z_t^u = j_d) \\
&= O_{i_1,j_1} \ldots O_{i_d,j_d} \\
&= (\bigotimes_{v \in H} O^v)_{(i_1,\ldots,i_d),(j_1,\ldots,j_d)}
\end{aligned}
$$

Where the second equality uses conditional independence. Therefore, $O^H = \bigotimes_{v \in H} O^v$. $\square$

### C.3.2 Application 2: HMM with rank-deficient observation matrix.

Consider an HMM whose observation matrix $O$ is rank-deficient. In this case, [3] suggests compressing sequences of successive observations of size $s$ for $s = 2, 3, \ldots$ until the matrices $\tilde{O}_s^f = P(x_t, x_{t+1}, \ldots, x_{t+s-1} | z_t)$ and $\tilde{O}_s^b = P(x_t, x_{t-1}, \ldots, x_{t-s+1} | z_t)$ have rank $m$. A version of [18] is then run using observation sequence pairs $P_{1:s,s+1:2s}$ and triples $P_{1:s,s+1,s+2:2s+1}$. In this case, we can show that both range($\tilde{O}_s^f$) and range($\tilde{O}_s^b$) are contained in range($O^{\otimes s}$); we can therefore use Product Projections to improve the $\Omega(n^s)$ running time to $O(m^{O(s)})$.

We first define forward and backward observation matrices $\tilde{O}_s^f$ and $\tilde{O}_s^b$ formally. For a fixed $s$, $\tilde{O}_s^f$ is a $n^s \times m$ matrix, with rows indexed by a $s$-tuple $(j_1, \ldots, j_s) \in [n]^s$, and columns indexed by $i \in [m]$. Entrywise,

$$(\tilde{O}_s^f)_{(i_1,\ldots,i_s),j} = P(x_t = i_1, x_{t+1} = i_2, \ldots, x_{t+s-1} = i_s | z_t = j)$$

Similarly we define backward observation matrices $\tilde{O}_s^b = P(x_t, x_{t-1}, \ldots, x_{t-s+1}|z_t)$. Entrywise,

$$(\tilde{O}_s^b)_{(i_1,\ldots,i_s),j} = P(x_t = i_1, x_{t-1} = i_2, \ldots, x_{t-s+1} = i_s | z_t = j)$$

The claim is the range of the forward(backward) observation matrices is contained in the range of the $s$-wise Kronecker product of the original observation matrices.

**Lemma 5.**
$$range(\tilde{O}_s^f) \subseteq range(O^{\otimes s})$$
$$range(\tilde{O}_s^b) \subseteq range(O^{\otimes s})$$

*Proof.* We prove the first relationship, since the proof of the second is almost identical. Note that by the law of total probability,

$$
\begin{aligned}
&(\tilde{O}_s^f)_{(i_1,i_2,\ldots,i_s),j} \\
=\ & P(x_t = i_1, x_{t+1} = i_2, \ldots, x_{t+s-1} = i_s | z_t = j) \\
=\ & \sum_{j_2,\ldots,j_s} P(x_t = i_1, x_{t+1} = i_2, \ldots, x_{t+s-1} = i_s | z_t = j, z_{t+1} = j_2, \ldots, z_{t+s-1} = j_s) \\
& \times P(z_{t+1} = j_2 \ldots, z_{t+s-1} = j_s | z_t = j) \\
=\ & \sum_{j_2,\ldots,j_s} O_{i_1,j} O_{i_2,j_2} \ldots O_{i_s,j_s} P(z_{t+1} = j_2 \ldots, z_{t+s-1} = j_s | z_t = j) \\
=\ & \sum_{j_2,\ldots,j_s} (O^{\otimes s})_{(i_1,i_2,\ldots,i_s),(j,j_2,\ldots,j_s)} P(z_{t+1} = j_2 \ldots, z_{t+s-1} = j_s | z_t = j)
\end{aligned}
$$

Thus, each column of $\tilde{O}_s^f$ is a linear combination of the columns of $O^{\otimes s}$, thus completing the proof.

$\square$

# D  Finite Sample Guarantees

**Theorem 2** (Accuracy of Initial Distribution and Transition Probabilities). *There exists a universal constant $C$ such that the following hold. Suppose Algorithm 1 is given as input $N$ iid observation triples $(x_{i1}, x_{i2}, x_{i3})_{i=1}^N$ generated by a THS-HMM, and outputs estimates of observaton matrices $\hat{O}^u$, for each node $u$ in the tree. Then Algorithm 2 is run on the same sample and has $\{\hat{O}^u\}_{u \in V}$ as input. If the size of sample $N$ is greater than:*

$$C \max\left( \frac{D^2}{\sigma_2^2 \sigma_3^2} \ln \frac{D}{\delta}, \frac{m}{\sigma_1^2 \sigma_2^2} \ln \frac{D}{\delta}, \frac{m^2}{\sigma_1^6 \sigma_3^6 \pi_{\min}^3} \ln \frac{D}{\delta}, \frac{m}{\sigma_2^2 \sigma_1^8 \epsilon^2} \ln \frac{D}{\delta}, \frac{m^2}{\sigma_3^6 \sigma_1^{14} \pi_{\min}^4 \epsilon^2} \ln \frac{D}{\delta} \right)$$

*where $\sigma_1 = \min_{u \in V} \sigma_m(O^u)$, $\sigma_2 = \min_{u \in V} \sigma_m(P_{1,2}^{u,u})$, $\sigma_3 = \min_{u \in V} \sigma_{m^d}(P_{1,3}^{H_u,H_u})$ and $\pi_{\min} = \min_{u,i} \pi_i^u$, then with probability $\geq 1 - \delta$ over the training examples, with probability $0.9$ over the random initializations in Algorithm 1, there exist permutation matrices $\{\Pi^u\}_{u \in V}$ such that for all $u \in V$,*

$$\|O^u - (\hat{O}^u \Pi^u)\| \leq \epsilon$$

*if $u$ is the root node, then,*

$$\|\hat{W}^u - (\Pi^u)^\top W^u\| \leq \epsilon$$
$$\|\hat{Q}^u - Q^u(\Pi^u, \Pi^u)\| \leq \epsilon$$

*Otherwise,*

$$\|\hat{W}^u - W^u(\Pi^u, \Pi^{\pi(u)})\| \leq \epsilon$$
$$\|\hat{Q}^u - Q^u(\Pi^u, \Pi^u, \Pi^{\pi(u)})\| \leq \epsilon$$

We emphasize that our algorithm recovers the initial probability and transition probability tensors up to permutations of hidden states in a *globally consistent* manner. In contrast to [20] where some hidden nodes do not have observations directly associated with them, in our setting, each hidden state has an associated observation, which makes recovery of permutations easier. How to perform parameter recovery in a THS-HMM with internal hidden states where each hidden tree node does not have an associated observation is an interesting question for future work.

# E  Proofs

Throughout this section, we first assume a technical condition on the sample size. This will result in concentration of the projection and the symmetrization matrices.

**Assumption 3.** *Recall that $D = |V|$. The sample size $N$ is large enough that*

$$
\begin{aligned}
&\epsilon(N, \delta) \\
&\leq \quad \min \Big( \frac{\min_{u \in V} \sigma_m(P_{1,2}^{u,u}) \min_{u \in V} \sigma_{m^d}(P_{1,3}^{H,H})}{16D}, \\
&\qquad \frac{\min_{u \in V} \sigma_m(P_{1,2}^{u,u}) \min_{u \in V} \sigma_m(O^u)}{4\sqrt{m}}, \frac{\min_{u \in V} \sigma_{m^d}(P_{1,3}^{H,H})^3 \min_{u \in V} \sigma_m(O^u)^3 \pi_{\min}^{3/2}}{1536 c_1 m} \Big) \\
&= \quad \min \Big( \frac{\sigma_2 \sigma_3}{16D}, \frac{\sigma_2 \sigma_1}{4\sqrt{m}}, \frac{\pi_{\min}^{3/2} \sigma_1^3 \sigma_3^3}{1536 c_1 m} \Big)
\end{aligned}
\tag{2}
$$

*Where $c_1 > 0$ is a constant given in Lemma 11, and $\sigma_1$, $\sigma_2$, $\sigma_3$ and $\pi_{\min}$ are defined in Theorem 2.*

## E.1  Raw Moments Concentration

We start with standard concentration of raw moments, which uses the fact that all the (vectorized) raw moments can be viewed as a probability vector. Let $u$ be a node in $V$, recall that $H$ is the set of nodes along the path from root $r$ to $u$.

Let $\epsilon(N, \delta) = \sqrt{\frac{1 + \ln(10D/\delta)}{N}}$. Define event

$$
\begin{aligned}
E = \Big\{ \text{ for all } u \in V : \quad & \|\hat{P}_{1,2}^{u,u} - P_{1,2}^{u,u}\|_F \leq \epsilon(N, \delta) \\
& \|\hat{P}_{1,2}^{H,u} - P_{1,2}^{H,u}\|_F \leq \epsilon(N, \delta) \\
& \|\hat{P}_{2,3}^{u,H} - P_{2,3}^{u,H}\|_F \leq \epsilon(N, \delta) \\
& \|\hat{P}_{1,3}^{H,H} - P_{1,3}^{H,H}\|_F \leq \epsilon(N, \delta) \\
& \|\hat{P}_{1,2,3}^{H,u,H} - P_{1,2,3}^{H,u,H}\|_F \leq \epsilon(N, \delta) \\
& \|\hat{P}_1^u - P_1^u\|_F \leq \epsilon(N, \delta) \\
& \|\hat{P}_{1,2}^{u,u} - P_{1,2}^{u,u}\|_F \leq \epsilon(N, \delta) \\
& \|\hat{P}_{1,1}^{u,\pi(u)} - P_{1,1}^{u,\pi(u)}\|_F \leq \epsilon(N, \delta) \\
& \|\hat{P}_{2,2,1}^{u,\pi(u),u} - P_{2,2,1}^{u,\pi(u),u}\|_F \leq \epsilon(N, \delta) \Big\}
\end{aligned}
$$

**Lemma 6** (Concentration of Raw Moments). $\mathbb{P}(E) \geq 1 - \delta$.

*Proof.* Applying Proposition 19 in [18] along with union bound. $\qquad \square$

## E.2  Subspace Concentration

Next we state a useful lemma that says that conditioned on the event $E$, performing an SVD on the empirical version of $P_{1,2}^{u,u} = \mathbb{E}[x_1^u \otimes x_2^u]$ gives us a good approximation to the range of $O^u$. Recall that $U^u$ is a matrix whose columns form an orthonormal basis of $O^u$, and define $U^H$ is $\otimes_{v \in H} U^u$. Also, recall for a matrix $U$ with orthonormal columns, the projection matrix onto range$(U)$ is $UU^\top$.

**Lemma 7** (Subspace Concentration). *Supposes $N$ is large enough such that Assumption 3 holds. $\hat{U}^u$ is the output of line 3 of Algorithm 1. Let $u$ be a node in $V$, recall that $H$ is the set of nodes along the path from root $r$ to $u$. Then conditioned on event $E$, we have:*
*(1)* $\|U^u(U^u)^\top - \hat{U}^u(\hat{U}^u)^\top\| \le \frac{2\epsilon(N,\delta)}{\sigma_m(P_{1,2}^{u,u})}$.

*In particular,*

$$\|U^u(U^u)^\top - \hat{U}^u(\hat{U}^u)^\top\| \le \min\left(\frac{\min_{u\in V}\sigma_m(P_{1,3}^{H,H})}{8D}, \frac{\min_{u\in V}\sigma_m(O^u)}{2\sqrt{m}}\right)$$

*(2)*

$$\|U^H(U^H)^\top - \hat{U}^H(\hat{U}^H)^\top\| \le \frac{\min_{u\in V}\sigma_m(P_{1,3}^{H_u,H_u})}{8}$$

*(3)*

$$\sigma_m((\hat{U}^u)^\top O^u) \ge \frac{\sigma_m(O^u)}{2}$$

*Proof.* (1) $\Phi^u$, the matrix of principal angles between range($\hat{U}^u$) and range($U^u$), is such that

$$\|\sin\Phi^u\|$$
$$\le \quad \frac{\epsilon(N,\delta)}{\sigma_m(P_{1,2}^{u,u}) - \epsilon(N,\delta)}$$
$$\le \quad \frac{2\epsilon(N,\delta)}{\sigma_m(P_{1,2}^{u,u})} \tag{3}$$

where the first inequality is by Theorem 4, by taking $A = P_{1,2}^{u,u}$ and $\tilde{A} = \hat{P}_{1,2}^{u,u}$; the second inequality from Assumption 3, which implies that $\epsilon(N,\delta) \le \sigma_m(P_{1,2}^{u,u})/2$.

Thus, by Equation (2) in Assumption 3,

$$\|\sin\Phi^u\| \le \min\left(\frac{\min_{u\in V}\sigma_m(P_{1,3}^{H_u,H_u})}{8D}, \frac{\min_{u\in V}\sigma_m(O^u)}{2\sqrt{m}}\right)$$

The result follows from the fact that

$$\|\sin\Phi^u\| = \|U^u(U^u)^\top - \hat{U}^u(\hat{U}^u)^\top\|$$

(2) First we enumerate the nodes in $H_u$ : $H_u = \{v_1,\ldots,v_l\}$.

$$\|U^H(U^H)^\top - \hat{U}^H(\hat{U}^H)^\top\|$$
$$\le \quad \|(U^{v_1}(U^{v_1})^\top - \hat{U}^{v_1}(\hat{U}^{v_1})^\top) \otimes \ldots \otimes (U^{v_l}(U^{v_l})^\top)|\| + \ldots + \|(U^{v_1}(U^{v_1})^\top) \otimes \ldots \otimes (U^{v_l}(U^{v_l})^\top - \hat{U}^{v_l}(\hat{U}^{v_l})^\top)\|$$
$$\le \quad \|U^{v_1}(U^{v_1})^\top - \hat{U}^{v_1}(\hat{U}^{v_1})^\top\| + \ldots + \|U^{v_l}(U^{v_l})^\top - \hat{U}^{v_l}(\hat{U}^{v_l})^\top\|$$
$$\le \quad \sum_{v\in H} \frac{2\epsilon(N,\delta)}{\sigma_m(P_{1,2}^{v,v})}$$
$$\le \quad \frac{\min_{u\in V}\sigma_m(P_{1,3}^{H,H})}{8}$$

where the first inequality is by triangle inequality, the second inequality uses standard facts about Kronecker product ($\|A\otimes B\| = \|A\|\|B\|$), the third inequality is from Equation (3), the fourth inequality is from Equation (2).

(3) By item (1) we know that

$$\|U^u(U^u)^\top - \hat{U}^u(\hat{U}^u)^\top\| \le \sigma_m(O^u)/(2\sqrt{m})$$

Hence

$$\|U^u(U^u)^\top O^u - \hat{U}^u(\hat{U}^u)^\top O^u\| \le \|U^u(U^u)^\top - \hat{U}^u(\hat{U}^u)^\top\|\|O^u\| \le \sigma_m(O^u)/2$$

where the second inequality is from the fact that $O^u$ is a column stochastic matrix, which implies that $\|O^u\| \le \|O^u\|_F \le \sqrt{m}$.
Therefore by Theorem 3,

$$\sigma_m((\hat{U}^u)^\top O^u) = \sigma_m(\hat{U}^u(\hat{U}^u)^\top O^u) \ge \sigma_m(O^u)/2$$

$\square$

### E.3 Symmetrized Moment Concentration

**Lemma 8.** *Suppose we are given a set of matrices $\hat{U}^u, u \in V$ such that $(\hat{U}^u)^\top O^u$ is invertible for all $u \in V$. Moreover, assume the expected second order moments $P_{2,3}^{u,H}, P_{1,3}^{H,H}, P_{2,1}^{u,H}$, and third order moments $P_{1,2,3}^{H,u,H}$ are given. Consider the symmetrization matrices:*

$$\tilde{S}_1^u = ((\hat{U}^u)^\top P_{2,3}^{u,H} \hat{U}^H)((\hat{U}^H)^\top P_{1,3}^{H,H} \hat{U}^H)^{-1}$$

$$\tilde{S}_3^u = ((\hat{U}^u)^\top P_{2,1}^{u,H} \hat{U}^H)((\hat{U}^H)^\top P_{3,1}^{H,H} \hat{U}^H)^{-1}$$

*and the ground truth symmetrized second order and third order cooccurence matrices be:*

$$M_2^u = P_{1,2}^{H,u}(\hat{U}^H(\tilde{S}_1^u)^\top, \hat{U}^u)$$

$$M_3^u = P_{1,2,3}^{H,u,H}(\hat{U}^H(\tilde{S}_1^u)^\top, \hat{U}^u, \hat{U}^H \tilde{S}_3^{uT})$$

*Then,*

$$M_2^u = \sum_i \pi_i^u ((\hat{U}^u)^\top O^u)_i \otimes ((\hat{U}^u)^\top O^u)_i$$

$$M_3^u = \sum_i \pi_i^u ((\hat{U}^u)^\top O^u)_i \otimes ((\hat{U}^u)^\top O^u)_i \otimes ((\hat{U}^u)^\top O^u)_i$$

*Proof.* Recall that by Lemma 2

$$O_1^H = O^H \mathrm{diag}(\rho^H)(T^H)^\top \mathrm{diag}(\pi^H)^{-1}$$

where $\mathrm{diag}(\rho^H)(T^H)^\top \mathrm{diag}(\pi^H)^{-1}$ is invertible. Thus,

$$(\hat{U}^H)^\top O_1^H = (\hat{U}^H)^\top O^H \mathrm{diag}(\rho^H)(T^H)^\top \mathrm{diag}(\pi^H)^{-1}$$

This shows that $(\hat{U}^H)^\top O_1^H$ is invertible.
On the other hand,

$$O_3^H = O^H T^H$$

where $T^H$ is invertible. Thus,

$$(\hat{U}^H)^\top O_3^H = (\hat{U}^H)^\top O^H T^H$$

This shows that $(\hat{U}^H)^\top O_3^H$ is invertible.

Therefore,

$$
\begin{aligned}
&\tilde{S}_1^u \\
=\ & ((\hat{U}^u)^\top O_2^u \mathrm{diag}(\pi^H) O_3^{HT} \hat{U}^H)((\hat{U}^H)^\top O_1^H \mathrm{diag}(\pi^H) O_3^{HT} \hat{U}^H)^{-1} \\
=\ & ((\hat{U}^u)^\top O_2^u)((\hat{U}^H)^\top O_1^H)^{-1}
\end{aligned}
$$

Likewise,

$$
\begin{aligned}
&\tilde{S}_3^u \\
=\ & ((\hat{U}^u)^\top O_2^u \mathrm{diag}(\pi^H) O_1^{HT} \hat{U}^H)((\hat{U}^H)^\top O_3^H \mathrm{diag}(\pi^H) O_1^{HT} \hat{U}^H)^{-1} \\
=\ & ((\hat{U}^u)^\top O_2^u)((\hat{U}^H)^\top O_3^H)^{-1}
\end{aligned}
$$

Then,

$$
\begin{aligned}
M_2^u &= P_{1,2}^{H,u}(U^H(\tilde{S}_1^u)^\top, \hat{U}^u) \\
&= \sum_{i_1,\ldots,i_D} \pi_{i_1,\ldots,i_D}^H ((\hat{U}^u)^\top O_2^u)_{i_1,\ldots,i_D} \otimes ((\hat{U}^u)^\top O_2^u)_{i_1,\ldots,i_D} \\
&= \sum_{i_1,\ldots,i_D} \pi_{i_1,\ldots,i_D}^H ((\hat{U}^u)^\top O^u)_{i_D} \otimes ((\hat{U}^u)^\top O^u)_{i_D} \\
&= \sum_i \pi_i^u ((\hat{U}^u)^\top O^u)_i \otimes ((\hat{U}^u)^\top O^u)_i
\end{aligned}
$$

$$
\begin{aligned}
M_3^u &= P_{1,2,3}^{H,u,H}(\hat{U}^H(\tilde{S}_1^u)^\top, \hat{U}^u, \hat{U}^H\tilde{S}_3^{uT}) \\
&= \sum_{i_1,\dots,i_D} \pi_{i_1,\dots,i_D}^H((\hat{U}^u)^\top O_2^u)_{i_1,\dots,i_D} \otimes ((\hat{U}^u)^\top O_2^u)_{i_1,\dots,i_D} \otimes ((\hat{U}^u)^\top O_2^u)_{i_1,\dots,i_D} \\
&= \sum_{i_1,\dots,i_D} \pi_{i_1,\dots,i_D}^H((\hat{U}^u)^\top O^u)_{i_D} \otimes ((\hat{U}^u)^\top O^u)_{i_D} \otimes ((\hat{U}^u)^\top O^u)_{i_D} \\
&= \sum_{i} \pi_i^u((\hat{U}^u)^\top O^u)_i \otimes ((\hat{U}^u)^\top O^u)_i \otimes ((\hat{U}^u)^\top O^u)_i
\end{aligned}
$$

$$\square$$

We next establish a result that shows that the symmetrization matrices $\hat{S}_1^u$ and $\hat{S}_3^u$ obtained in Line 7 of Algorithm 1 concentrate to $\tilde{S}_1^u$ and $\tilde{S}_3^u$ defined in Lemma 8. Recall from Algorithm 1 that:

$$
\hat{S}_1^u = ((\hat{U}^u)^\top \hat{P}_{2,3}^{u,H_u}\hat{U}^{H_u})((\hat{U}^{H_u})^\top \hat{P}_{1,3}^{H_u,H_u}\hat{U}^{H_u})^{-1}, \quad \hat{S}_3^u = ((\hat{U}^u)^\top \hat{P}_{2,1}^{u,H_u}\hat{U}^{H_u})((\hat{U}^{H_u})^\top \hat{P}_{3,1}^{H_u,H_u}\hat{U}^{H_u})^{-1}
$$

**Lemma 9.** *Suppose $N$ is large enough that Assumption 3 holds. Recall $\hat{S}_1^u$ and $\hat{S}_3^u$ are the outputs of line 7 in Algorithm 1 , and $\tilde{S}_1^u$ and $\tilde{S}_3^u$ are defined in Lemma 8. Conditioned on event E, the following hold for all $u \in V$.*

$$
\|\tilde{S}_1^u - \hat{S}_1^u\|, \|\tilde{S}_3^u - \hat{S}_3^u\| \le \frac{10\epsilon(N,\delta)}{\sigma_{m^d}(P_{1,3}^{H,H})^2}
$$

$$
\|\tilde{S}_1^u\|, \|\hat{S}_1^u\|, \|\tilde{S}_3^u\|, \|\hat{S}_3^u\| \le \frac{4}{\sigma_{m^d}(P_{1,3}^{H,H})}
$$

*Proof.* (1) We first show that $\sigma_{m^d}((\hat{U}^H)^\top \hat{P}_{1,3}^{H,H}\hat{U}^H) \ge 3\sigma_{m^d}(P_{1,3}^{H,H})/4$, and $\sigma_{m^d}((\hat{U}^H)^\top P_{1,3}^{H,H}\hat{U}^H) \ge \sigma_{m^d}(P_{1,3}^{H,H})/2$.
Under Assumption 3, by Item (2) of Lemma 7, we know that

$$
\|U^H(U^H)^\top - \hat{U}^H(\hat{U}^H)^\top\| \le \min_{u\in V} \sigma_{m^d}(P_{1,3}^{H,H})/8 \tag{4}
$$

As a result,

$$
\begin{aligned}
&\|\hat{U}^H(\hat{U}^H)^\top P_{1,3}^{H,H}\hat{U}^H(\hat{U}^H)^\top - P_{1,3}^{H,H}\| \\
=\ & \|\hat{U}^H(\hat{U}^H)^\top P_{1,3}^{H,H}\hat{U}^H(\hat{U}^H)^\top - U^H(U^H)^\top P_{1,3}^{H,H}U^H(U^H)^\top\| && (5) \\
\le\ & \|(\hat{U}^H(\hat{U}^H)^\top - U^H(U^H)^\top)P_{1,3}^{H,H}\hat{U}^H(\hat{U}^H)^\top\| \\
& + \|U^H(U^H)^\top P_{1,3}^{H,H}(\hat{U}^H(\hat{U}^H)^\top - U^H(U^H)^\top)\| \\
\le\ & \|(\hat{U}^H(\hat{U}^H)^\top - U^H(U^H)^\top)\|\|P_{1,3}^{H,H}\|\|\hat{U}^H(\hat{U}^H)^\top\| \\
& + \|U^H(U^H)^\top\|\|P_{1,3}^{H,H}\|\|(\hat{U}^H(\hat{U}^H)^\top - U^H(U^H)^\top)\| \\
\le\ & \sigma_m(P_{1,3}^{H,H})/8 + \sigma_m(P_{1,3}^{H,H})/8 \\
\le\ & \sigma_m(P_{1,3}^{H,H})/4 && (6)
\end{aligned}
$$

where the first inequality is by triangle inequality, in the second inequality we use the fact that $\|A \cdot B\| \le \|A\|\|B\|$, the third inequality is from the fact that $\|P_{1,3}^{H,H}\| \le \|P_{1,3}^{H,H}\|_F \le 1$, $\|\hat{U}^H(\hat{U}^H)^\top\| = 1$, $\|U^H(U^H)^\top\| = 1$ and Equation (4).
Therefore,

$$
\begin{aligned}
& \sigma_{m^d}((\hat{U}^H)^\top P_{1,3}^{H,H}\hat{U}^H) \\
=\ & \sigma_{m^d}(\hat{U}^H(\hat{U}^H)^\top P_{1,3}^{H,H}\hat{U}^H(\hat{U}^H)^\top) \\
\ge\ & \sigma_{m^d}(P_{1,3}^{H,H}) - \|\hat{U}^H(\hat{U}^H)^\top P_{1,3}^{H,H}\hat{U}^H(\hat{U}^H)^\top - P_{1,3}^{H,H}\| \\
\ge\ & 3\sigma_{m^d}(P_{1,3}^{H,H})/4 && (7)
\end{aligned}
$$

where the first inequality is by Theorem 3, the second inequality is by Equation 6.
In the meantime,

$$
\begin{aligned}
&\|(\hat{U}^H)^\top P_{1,3}^{H,H}\hat{U}^H - (\hat{U}^H)^\top \hat{P}_{1,3}^{H,H}\hat{U}^H\| \\
&\leq \quad \|P_{1,3}^{H,H} - \hat{P}_{1,3}^{H,H}\| \\
&\leq \quad \epsilon(N,\delta) \leq \sigma_m^d(P_{1,3}^{H,H})/4
\end{aligned}
\tag{8}
$$

where in the first inequality we use the fact that $\|\hat{U}^H\| = 1$, the second inequality is by the fact that if $E$ happens, $\|P_{1,3}^{H,H} - \hat{P}_{1,3}^{H,H}\| \leq \epsilon(N,\delta)$, the third inequality follows from Assumption 3.

Therefore

$$
\begin{aligned}
&\sigma_{m^d}((\hat{U}^H)^\top \hat{P}_{1,3}^{H,H}\hat{U}^H) \\
&\geq \quad \sigma_{m^d}((\hat{U}^H)^\top P_{1,3}^{H,H}\hat{U}^H) - \|(\hat{U}^H)^\top P_{1,3}^{H,H}\hat{U}^H - (\hat{U}^H)^\top \hat{P}_{1,3}^{H,H}\hat{U}^H\| \\
&\geq \quad \sigma_m(P_{1,3}^{H,H})/2
\end{aligned}
$$

where the first inequality is from Theorem 3, the second inequality is from Equation (8).

We now have

$$
\begin{aligned}
&\|\tilde{S}_1^u - \hat{S}_1^u\| \\
&= \quad \|((\hat{U}^u)^\top P_{2,3}^{u,H}\hat{U}^H)((\hat{U}^H)^\top P_{1,3}^{H,H}\hat{U}^H)^{-1} - ((\hat{U}^u)^\top \hat{P}_{2,3}^{u,H}\hat{U}^H)((\hat{U}^H)^\top \hat{P}_{1,3}^{H,H}\hat{U}^H)^{-1}\| \\
&\leq \quad \|((\hat{U}^u)^\top (P_{2,3}^{u,H} - \hat{P}_{2,3}^{u,H})\hat{U}^H)((\hat{U}^H)^\top P_{1,3}^{H,H}\hat{U}^H)^{-1}\| \\
&\quad + \|((\hat{U}^u)^\top \hat{P}_{2,3}^{u,H}\hat{U}^H)(((\hat{U}^H)^\top P_{1,3}^{H,H}\hat{U}^H)^{-1} - ((\hat{U}^H)^\top \hat{P}_{1,3}^{H,H}\hat{U}^H)^{-1})\| \\
&\leq \quad \|\hat{U}^u(P_{2,3}^{u,H} - \hat{P}_{2,3}^{u,H})\hat{U}^H\|\|((\hat{U}^H)^\top P_{1,3}^{H,H}\hat{U}^H)^{-1}\| \\
&\quad + \|(\hat{U}^u)^\top \hat{P}_{2,3}^{u,H}\hat{U}^H\|\|((\hat{U}^H)^\top P_{1,3}^{H,H}\hat{U}^H)^{-1} - ((\hat{U}^H)^\top \hat{P}_{1,3}^{H,H}\hat{U}^H)^{-1}\| \\
&\leq \quad \frac{2\epsilon(N,\delta)}{\sigma_{m^d}(P_{1,3}^{H,H})} + \frac{8\epsilon(N,\delta)}{\sigma_{m^d}(P_{1,3}^{H,H})^2} \\
&\leq \quad \frac{10\epsilon(N,\delta)}{\sigma_{m^d}(P_{1,3}^{H,H})^2}
\end{aligned}
\tag{9}
$$

In the derivation above, the first inequality uses triangle inequality and the second inequality repeatedly uses the fact that $\|A \cdot B\| \leq \|A\|\|B\|$. The third inequality is obtained by bounding each term individually as follows:

$$
\begin{aligned}
&\|(\hat{U}^u)^\top (P_{2,3}^{u,H} - \hat{P}_{2,3}^{u,H})\hat{U}^H\| \leq \|P_{2,3}^{u,H} - \hat{P}_{2,3}^{u,H}\| \leq \|P_{2,3}^{u,H} - \hat{P}_{2,3}^{u,H}\|_F \leq \epsilon(N,\delta) \\
&\|((\hat{U}^H)^\top P_{1,3}^{H,H}\hat{U}^H)^{-1}\| = 1/\sigma_{m^d}((\hat{U}^H)^\top P_{1,3}^{H,H}\hat{U}^H) \leq 2/\sigma_{m^d}(P_{1,3}^{H,H}) \\
&\|(\hat{U}^u)^\top \hat{P}_{2,3}^{u,H}\hat{U}^H\| \leq \|\hat{P}_{2,3}^{u,H}\| \leq \|\hat{P}_{2,3}^{u,H}\|_F \leq 1 \\
&\|((\hat{U}^H)^\top P_{1,3}^{H,H}\hat{U}^H)^{-1} - ((\hat{U}^H)^\top \hat{P}_{1,3}^{H,H}\hat{U}^H)^{-1}\| \\
&\leq \quad 2\|(\hat{U}^H)^\top (P_{1,3}^{H,H} - \hat{P}_{1,3}^{H,H})\hat{U}^H\| \max(\|((\hat{U}^H)^\top \hat{P}_{1,3}^{H,H}\hat{U}^H)^{-1}\|, \|((\hat{U}^H)^\top P_{1,3}^{H,H}\hat{U}^H)^{-1}\|) \\
&\leq \quad \frac{8\epsilon(N,\delta)}{\sigma_{m^d}(P_{1,3}^{H,H})^2}
\end{aligned}
$$

where the last inequality follows from Theorem 5.
The bound of $\|\tilde{S}_3^u - \hat{S}_3^u\|$ is handled similarly.

(2) First,

$$
\|\tilde{S}_1^u\| \leq \|(\hat{U}^u)^\top P_{2,1}^{u,H}\hat{U}^H\|\|((\hat{U}^H)^\top P_{3,1}^{H,H}\hat{U}^H)^{-1}\| \leq \frac{2}{\sigma_{m^d}(P_{1,3}^{H,H})}
$$

where the first inequality is by the fact that $\|A \cdot B\| \leq \|A\|\|B\|$, the second inequality is by Equation (7).

Meanwhile, Assumption 3 implies $\epsilon(N, \delta) \le \sigma_{m^d}(P_{1,3}^{H,H})/5$, therefore from Equation (9),

$$\|S_1^u - \hat{S}_1^u\| \le \frac{2}{\sigma_{m^d}(P_{1,3}^{H,H})}$$

Hence by triangle inequality,

$$\|\hat{S}_1^u\| \le \frac{4}{\sigma_{m^d}(P_{1,3}^{H,H})}$$

The bounds of $\|\tilde{S}_3^u\|$ and $\|\hat{S}_3^u\|$ are handled similarly. $\qquad\square$

Built upon the previous two lemmas, we next provide a result regarding the concentration of symmetrized moments.

**Lemma 10.** *Suppose $N$ is large enough that Assumption 3 holds. Let $u$ be a node in $V$. Then on the event $E$, the following hold.*

$$\|M_2^u - \hat{M}_2^u\| \le \frac{14\epsilon(N, \delta)}{\sigma_{m^d}(P_{1,3}^{H,H})^2}$$

$$\|M_3^u - \hat{M}_3^u\| \le \frac{96\epsilon(N, \delta)}{\sigma_{m^d}(P_{1,3}^{H,H})^3}$$

*Proof.* (1) Define $P^u = P_{1,2}^{H,u}(\hat{U}^H, \hat{U}^u)$ and $\hat{P}^u = \hat{P}_{1,2}^{H,u}(\hat{U}^H, \hat{U}^u)$. Then,

$$
\begin{aligned}
&\|M_2^u - \hat{M}_2^u\| \\
=\quad & \|P^u((\tilde{S}_1^u)^\top, I) - \hat{P}^u((\hat{S}_1^u)^\top, I)\| \\
\le\quad & \|(P^u - \hat{P}^u)((\tilde{S}_1^u)^\top, I)\| + \|\hat{P}^u((\tilde{S}_1^u)^\top - (\hat{S}_1^u)^\top, I)\| \\
\le\quad & \|P^u - \hat{P}^u\|\|\tilde{S}_1^u\| + \|\hat{P}^u\|\|\tilde{S}_1^u - \hat{S}_1^u\| \\
\le\quad & \frac{4\epsilon(N, \delta)}{\sigma_{m^d}(P_{1,3}^{H,H})} + \frac{10\epsilon(N, \delta)}{\sigma_{m^d}(P_{1,3}^{H,H})^2} \le \frac{14\epsilon(N, \delta)}{\sigma_{m^d}(P_{1,3}^{H,H})^2} \qquad (10)
\end{aligned}
$$

where the first inequality is by triangle inequality, the second inequality is by the fact that $\|M(A, B)\| \le \|M\|\|A\|\|B\|$, the third inequality is from the fact that $\|P^u - \hat{P}^u\| \le \|P_{1,2}^{H,u} - \hat{P}_{1,2}^{H,u}\| \le \|P_{1,2}^{H,u} - \hat{P}_{1,2}^{H,u}\|_F \le \epsilon(N, \delta)$ and $\|\hat{P}^u\| \le \|\hat{P}_{1,2}^{H,u}\| \le \|\hat{P}_{1,2}^{H,u}\|_F \le 1$, and Lemma 9.

As a result,

$$
\begin{aligned}
&\|M_2^u - \hat{M}_2^u\| \\
=\quad & \|(P^u((\tilde{S}_1^u)^\top, I)^\top + P^u((\tilde{S}_1^u)^\top, I))/2 - (\hat{P}^u(\hat{S}_1^u, I)^\top + \hat{P}^u(\hat{S}_1^u, I))/2\| \\
\le\quad & \|P^u((\tilde{S}_1^u)^\top, I)^\top - \hat{P}^u((\hat{S}_1^u)^\top, I)^\top\|/2 + \|P^u((\tilde{S}_1^u)^\top, I) - \hat{P}^u((\hat{S}_1^u)^\top, I)\|/2 \\
\le\quad & \frac{14\epsilon(N, \delta)}{\sigma_{m^d}(P_{1,3}^{H,H})^2}
\end{aligned}
$$

where the first inequality follows from triangle inequality, the second inequality is from Equation (10).

(2) Define $T^u = P_{1,2,3}^{H,u,H}(\hat{U}^H, \hat{U}^u, \hat{U}^H)$ and $\hat{T}^u = \hat{P}_{1,2,3}^{H,u,H}(\hat{U}^H, \hat{U}^u, \hat{U}^H)$. Then,

$$
\begin{aligned}
&\|M_3^u - \hat{M}_3^u\| \\
=\quad & \|T^u((\tilde{S}_1^u)^\top, I, (\tilde{S}_3^u)^\top) - \hat{T}^u((\hat{S}_1^u)^\top, I, (\hat{S}_3^u)^\top)\| \\
\le\quad & \|T^u - \hat{T}^u\|\|\tilde{S}_1^u\|\|\tilde{S}_3^u\| + \|\hat{T}^u\|\|\tilde{S}_1^u - \hat{S}_1^u\|\|\tilde{S}_3^u\| + \|\hat{T}^u\|\|\hat{S}_1^u\|\|\tilde{S}_3^u - \hat{S}_3^u\| \\
\le\quad & \frac{16\epsilon(N, \delta)}{\sigma_{m^d}(P_{1,3}^{H,H})^2} + \frac{10\epsilon(N, \delta)}{\sigma_{m^d}(P_{1,3}^{H,H})^2}\frac{4}{\sigma_{m^d}(P_{1,3}^{H,H})} + \frac{4}{\sigma_{m^d}(P_{1,3}^{H,H})}\frac{10\epsilon(N, \delta)}{\sigma_{m^d}(P_{1,3}^{H,H})^2} \\
\le\quad & \frac{96\epsilon(N, \delta)}{\sigma_{m^d}(P_{1,3}^{H,H})^3}
\end{aligned}
$$

where the first inequality is from triangle inequality, and the fact that $\|T(A, B, C)\| \leq \|T\|\|A\|\|B\|\|C\|$, the second inequality is by the fact that $\|T^u - \hat{T}^u\| \leq \|P_{1,2,3}^{H,u,H} - \hat{P}_{1,2,3}^{H,u,H}\| \leq \|P_{1,2,3}^{H,u,H} - \hat{P}_{1,2,3}^{H,u,H}\|_F \leq \epsilon(N, \delta)$, $\|\hat{T}^u\| \leq \|\hat{P}_{1,2,3}^{H,u,H}\| \leq 1$, and Lemma 9, the third inequality is by algebra. $\square$

## E.4 Accucary of Tensor Decomposition

---

**Algorithm 3** A Procedure That Finds Symmetric Decomposition based on Second and Third Order Moments

---

1: Input: number of components $m$, perturbed version $\hat{M}_2$ and $\hat{M}_3$ of matrix $M_2$ and tensor $M_3$ satisfying $M_2 = \sum_{i=1}^{m} \pi_i \theta_i \otimes \theta_i$, $M_3 = \sum_{i=1}^{m} \pi_i \theta_i \otimes \theta_i \otimes \theta_i$
2: Output: $\{\hat{\theta}_i\}_{i=1}^{m}$, estimate of $\{\theta_i\}_{i=1}^{m}$
3: **Whiten.** Perform an SVD on $\hat{M}_2 = \hat{U}\hat{D}\hat{U}^\top$, and let $\hat{W} = \hat{U}_m \hat{D}_m^{-1/2}$(where $\hat{U}_m$ is matrix that contains the first $m$ columns of $\hat{U}$, $\hat{D}_m$ is the diagonal matrix with $\hat{D}$'s first $m$ diagonal entries), let $\hat{G} = \hat{M}_3(\hat{W}, \hat{W}, \hat{W})$.
4: **Decompose Tensor.** Apply robust tensor power iteration algorithm in [1] with input $\hat{G}$ to get $\{\hat{v}_1, \ldots, \hat{v}_m\}$
5: **for** $i = 1, 2, \ldots, m$ **do**
6:    Let $\hat{Z}_i = \frac{1}{\hat{T}(\hat{v}_i, \hat{v}_i, \hat{v}_i)}$.
7:    Recover $\hat{\theta}_i = \frac{(\hat{W}^\top)^\dagger \hat{v}_i}{\hat{Z}_i}$
8: **end for**

---

In this section, we introduce a lemma that is implicit in [1] regarding using orthogonal decomposition as a subprocedure for full rank symmetric tensor decomposition. (See Theorem 5.1 of [1].) For completeness, we include the proof here.

**Lemma 11.** *There are universal constants $c_1$, $c_2$ such that the following holds. Suppose a matrix $M_2$ and a tensor $M_3$ has the following structure:*

$$M_2 = \sum_{i=1}^{m} \pi_i \theta_i \otimes \theta_i$$

$$M_3 = \sum_{i=1}^{m} \pi_i \theta_i \otimes \theta_i \otimes \theta_i$$

*where $\pi_i > 0$ for all $i$. And we are given their perturbed version $\hat{M}_2$ and $\hat{M}_3$, such that*

$$\|\hat{M}_2 - M_2\| \leq E_P$$

$$\|\hat{M}_3 - M_3\| \leq E_T$$

*where*

$$E_P \leq \sigma_m(\Theta)^2 \pi_{\min}/2 \tag{11}$$

$$c_1 \left( \frac{E_T}{\sigma_m(\Theta)^3} + \frac{E_P}{\sigma_m(\Theta)^2} \right) \frac{1}{\pi_{\min}^{3/2}} \leq \frac{1}{m} \tag{12}$$

*where $\Theta = (\theta_1, \ldots \theta_m)$ and $\pi_{\min} = \min_i \pi_i$. Then the outputs $\{\theta_i\}_{i=1}^{m}$ of Algorithm 3 on input $\hat{M}_2$ and $\hat{M}_3$ satisfies the following. With appropriate setting of parameters (with respect to parameter $\eta$), with probability $1 - \eta$, there is a permutation $\sigma : [m] \rightarrow [m]$ such that*

$$\|\theta_i - \hat{\theta}_{\sigma(i)}\| \leq c_2 \frac{\sigma_1(\Theta)}{\pi_{\min}^2} \left( \frac{E_P}{\sigma_m(\Theta)^2} + \frac{E_T}{\sigma_m(\Theta)^3} \right)$$

*Proof.* 1. We first put $\Theta$ into canonical forms by appropriate scaling of its columns. Let $\tilde{\Theta} = (\tilde{\theta}_1, \ldots, \tilde{\theta}_m) = \Theta \text{diag}(\pi)^{\frac{1}{2}}$, we have

$$M_2 = \sum_{i=1}^m \tilde{\theta}_i \otimes \tilde{\theta}_i$$

$$M_3 = \sum_{i=1}^m \frac{1}{\sqrt{\pi_i}} \tilde{\theta}_i \otimes \tilde{\theta}_i \otimes \tilde{\theta}_i$$

Recall that $\hat{W}$ is defined as $\hat{U}_m \hat{D}_m^{-\frac{1}{2}}$, where $\hat{M}_2 = \hat{U}\hat{D}\hat{U}^\top$. Hence $\hat{W}^\top \hat{M}_2 \hat{W} = I_m$. Suppose that $\hat{W}^\top M_2 \hat{W}$ has the following eigendecomposition:

$$\hat{W}^\top M_2 \hat{W} = A\Lambda A^\top$$

Then let $W = \hat{W} A \Lambda^{-\frac{1}{2}} A^\top$, $W$ is one of the matrices such that $W^\top M_2 W = I_m$. Define $M = W^\top \tilde{\Theta}$, $\hat{M} = \hat{W}^\top \tilde{\Theta}$.

2. If Equation (11) holds, then $E_p \le \sigma_m(\Theta)^2 \pi_{\min}/2 \le \sigma_m(M_2)/2$, then we have the following:

$$\|W\|, \|\hat{W}\| \le \frac{2}{\sigma_m(\tilde{\Theta})}$$

$$\|W^\dagger\|, \|\hat{W}^\dagger\| \le 3\sigma_1(\tilde{\Theta})$$

$$\|W^\dagger - \hat{W}^\dagger\| \le \frac{6\sigma_1(\tilde{\Theta})}{\sigma_m(\tilde{\Theta})^2} E_P$$

$$\|\Theta\Theta^\dagger - WW^\dagger\| \le \frac{4E_P}{\sigma_m(\tilde{\Theta})}$$

$$\|M\|, \|\hat{M}\| \le 2$$

$$\|M - \hat{M}\| \le \frac{E_P}{\sigma_m(\tilde{\Theta})^2}$$

3. Define $G = M_3(W, W, W) = \sum_i \frac{1}{\sqrt{\pi_i}} M_i \otimes M_i \otimes M_i$, and recall that $\hat{G} = \hat{M}_3(\hat{W}, \hat{W}, \hat{W})$. We have the following perturbation bound for $\hat{G}$. Define $R$ to be diagonal tensor $\sum_i \frac{1}{\sqrt{\pi_i}} e_i \otimes e_i \otimes e_i$. Note that $\|R\| \le \frac{1}{\sqrt{\pi_{\min}}}$. Therefore,

$$
\begin{aligned}
&\|G - \hat{G}\| \\
=\ &\|M_3(W, W, W) - \hat{M}_3(\hat{W}, \hat{W}, \hat{W})\| \\
\le\ &\|(M_3 - \hat{M}_3)(\hat{W}, \hat{W}, \hat{W})\| + \|M_3(W - \hat{W}, W, W)\| + \|M_3(\hat{W}, W - \hat{W}, W)\| + \|M_3(\hat{W}, \hat{W}, W - \hat{W})\| \\
=\ &\|(M_3 - \hat{M}_3)(\hat{W}, \hat{W}, \hat{W})\| + \|R(M - \hat{M}, M, M)\| + \|R(\hat{M}, M - \hat{M}, M)\| + \|R(\hat{M}, \hat{M}, M - \hat{M})\| \\
\le\ &\|M_3 - \hat{M}_3\|\|W\|^3 + \|R\|\|M - \hat{M}\|\|M\|^2 + \|R\|\|\hat{M}\|\|M\|\|M - \hat{M}\| + \|R\|\|\hat{M}\|^2\|M - \hat{M}\| \\
\le\ &\frac{8E_T}{\sigma_m(\tilde{\Theta})^3} + \frac{12E_P}{\sqrt{\pi_{\min}}\sigma_m(\tilde{\Theta})^2} := E
\end{aligned}
\tag{13}
$$

where the first inequality is by triangle inequality, the second inequality is by the fact that $\|T(A, B, C)\| \le \|T\|\|A\|\|B\|\|C\|$, the third inequality is from results of our step 2 and the fact that $\|\hat{M}_3 - M_3\| \le E_T$.

4. If Equation (12) holds, then $E \le \frac{C_1}{m} \le C_1 \frac{\min_i \pi_i^{-1/2}}{m}$ for $C_1$ required by Theorem 5.1 in [1]. Thus, applying robust tensor power algorithm in [1], with probability at least $1 - \eta$, there exist a permutation $\sigma : [m] \to [m]$ such that

$$\|M_i - \hat{v}_{\sigma(i)}\| \le 8\sqrt{\pi_i} E \tag{14}$$

5. We conclude by providing the reconstruction error bound. For notational simplicity, assume $\sigma(\cdot)$ is identity mapping. Define

$$Z_i = \frac{1}{M_3(WM_i, WM_i, WM_i)} = \frac{1}{G(M_i, M_i, M_i)} = \sqrt{\pi_i}$$

and recall that

$$\hat{Z}_i = \frac{1}{\hat{M}_3(\hat{W}\hat{v}_i, \hat{W}\hat{v}_i, \hat{W}\hat{v}_i)} = \frac{1}{\hat{G}(\hat{v}_i, \hat{v}_i, \hat{v}_i)}$$

The recovery formula is

$$\hat{\theta}_i = \frac{(\hat{W}^\top)^\dagger \hat{v}_i}{\hat{Z}_i}$$

First, $|\frac{1}{Z_i} - \frac{1}{\hat{Z}_i}|$ can be bounded as follows:

$$
\begin{aligned}
&|\frac{1}{Z_i} - \frac{1}{\hat{Z}_i}| \\
=\ & |G(M_i, M_i, M_i) - \hat{G}(\hat{v}_i, \hat{v}_i, \hat{v}_i)| \\
\leq\ & |(G - \hat{G})(\hat{v}_i, \hat{v}_i, \hat{v}_i)| + |G(M_i - \hat{v}_i, \hat{v}_i, \hat{v}_i)| + |G(M_i, M_i - \hat{v}_i, \hat{v}_i)| + |G(M_i, M_i, M_i - \hat{v}_i)| \\
\leq\ & \|G - \hat{G}\|\|M_i\|^3 + \|G\|\|M_i - \hat{v}_i\|\|\hat{v}_i\|^2 + \|G\|\|M_i - \hat{v}_i\|\|\hat{v}_i\|\|M_i\| + \|G\|\|M_i\|^2\|M_i - \hat{v}_i\| \\
\leq\ & E + 3\frac{\pi_i}{\sqrt{\pi_{\min}}} E \leq 4\frac{\pi_i}{\sqrt{\pi_{\min}}} E
\end{aligned}
$$

where the first inequality is by triangle inequality, the second inequality is by the fact that $\|A \cdot B\| \leq \|A\|\|B\|$, the third inequality is by Equation (13) in step 3 and Equation (14) in step 4, the fourth inequality is by algebra.

Then the reconstruction error can be bounded as follows:

$$
\begin{aligned}
&\|\theta_i - \frac{(\hat{W}^\dagger)^\top \hat{v}_i}{\hat{Z}_i}\| \\
\leq\ & \|\theta_i - \frac{(W^\dagger)^\top M_i}{Z_i}\| + \|\frac{W^\dagger(M_i - \hat{v}_i)}{Z_i}\| + \|\frac{(W^\dagger - \hat{W}^\dagger)\hat{v}_i}{Z_i}\| + \|(\frac{1}{Z_i} - \frac{1}{\hat{Z}_i})\hat{W}^\dagger \hat{v}_i\| \\
\leq\ & \|\Theta\Theta^\dagger - WW^\dagger\|\|\theta_i\| + \frac{\|W^\dagger\|}{Z_i}\|M_i - \hat{v}_i\| + \frac{\|W^\dagger - \hat{W}^\dagger\|}{Z_i}\|\hat{v}_i\| + |\frac{1}{Z_i} - \frac{1}{\hat{Z}_i}|\|\hat{W}^\dagger\|\|\hat{v}_i\| \\
\leq\ & \|\Theta\Theta^\dagger - WW^\dagger\|\frac{\sigma_1(\tilde{\Theta})}{\sqrt{\pi_{\min}}} + \frac{\|W^\dagger\|}{Z_i}\|M_i - \hat{v}_i\| + \frac{\|W^\dagger - \hat{W}^\dagger\|}{Z_i} + |\frac{1}{Z_i} - \frac{1}{\hat{Z}_i}|\|\hat{W}^\dagger\| \\
\leq\ & \frac{4E_P}{\sigma_m(\tilde{\Theta})^2}\frac{\sigma_1(\tilde{\Theta})}{\sqrt{\pi_{\min}}} + 24\sigma_1(\tilde{\Theta})E + \frac{6\sigma_1(\tilde{\Theta})}{\sigma_m(\tilde{\Theta})^2}E_P\sqrt{\pi_i} + \frac{12}{\sqrt{\pi_{\min}}}E\sqrt{\pi_i}\sigma_1(\tilde{\Theta}) \\
\leq\ & \frac{46\sigma_1(\tilde{\Theta})}{\sqrt{\pi_{\min}}}(\frac{8E_T}{\sigma_m(\tilde{\Theta})^3} + \frac{12E_P}{\sqrt{\pi_{\min}}\sigma_m(\tilde{\Theta})^2}) \\
\leq\ & c_2\frac{\sigma_1(\Theta)}{\pi_{\min}^2}(\frac{E_P}{\sigma_m(\Theta)^2} + \frac{E_T}{\sigma_m(\Theta)^3})
\end{aligned}
$$

Wher the first inequality is by triangle inequality, the second inequality we use the fact that $\|A \cdot B\| \leq \|A\|\|B\|$ and the fact that $M_i = W^\top \theta_i \sqrt{\pi_i}$, $Z_i = \sqrt{\pi_i}$, $\Theta\Theta^\dagger\theta_i = \theta_i$, $WW^\dagger = (W^\dagger)^\top W^\top$, the third inequality uses the fact that $\|\theta_i\| = \|\tilde{\Theta}e_i\|/\sqrt{\pi_{\min}} \leq \sigma_1(\tilde{\Theta})/\sqrt{\pi_{\min}}$ and $\|\hat{v}_i\| = 1$, in the fourth inequality we use results in item 2 and item 4, the fifth inequality is from the definiton of $E$ and algebra, in the sixth inequality we use the fact that $\sigma_m(\Theta) \leq \sigma_m(\tilde{\Theta})\pi_{\min}^{-1/2}$ and letting $c_2 = 552$. □

Now we apply the above lemma into our symmetrized cooccurence matrices $\hat{M}_2$ and $\hat{M}_3$.

**Corollary 2.** *Suppose $N$ is large enough such that Assumption 3 holds. Then, on event $E$, with probability $0.9$ over the randomization of $D$ calls of Algorithm 3, for all $u \in V$, the matrices $\hat{\Theta}^u = (\hat{\theta}_1^u, \ldots, \hat{\theta}_m^u)$ obtained at the end of line 9 are such that there exists a permutation matrix $\Pi^u$,*

$$\|(\hat{U}^u)^\top O^u - \hat{\Theta}^u\Pi^u\| \leq 2c_2\frac{m}{(\pi_{\min}^u)^2}\frac{\epsilon(N,\delta)}{\sigma_{m^d}(P_{1,3}^{H,H})^3\sigma_m(O^u)^3}$$

*Proof.* By Assumption 3, we first see that conditioned on event $E$, by Lemma 9, $\sigma_m(\hat{U}^{uT}O^u) \geq \sigma_m(O^u)/2$. Thus the conditions of Lemma 11 hold, by taking $\Theta = (\hat{U}^u)^\top O^u$, $\pi = \pi^u$. We thus get that with probability greater than $1 - 0.1/D$ over the randomness of Algorithm 1, there is a permutation matrix $\Pi^u$ such that for all $i = 1, 2, \ldots, m$,

$$
\begin{aligned}
&\|(\hat{U}^u)^\top O_i^u - (\hat{\Theta}^u \Pi^u)_i\| \\
&\leq\quad c_2 \frac{\sigma_1((\hat{U}^u)^\top O^u)}{(\pi_{\min}^u)^2} \left( \frac{\epsilon(N, \delta)}{\sigma_{m^d}(P_{1,3}^{H,H})^2 \sigma_m(O^u)^2} + \frac{\epsilon(N, \delta)}{\sigma_{m^d}(P_{1,3}^{H,H})^3 \sigma_m(O^u)^3} \right) \\
&\leq\quad 2c_2 \frac{\sqrt{m}}{(\pi_{\min}^u)^2} \frac{\epsilon(N, \delta)}{\sigma_{m^d}(P_{1,3}^{H,H})^3 \sigma_m(O^u)^3}
\end{aligned}
$$

where the second inequality we use the fact that $\sigma_1((\hat{U}^u)^\top O^u) = \|(\hat{U}^u)^\top O^u\| \leq \|O^u\| \leq \sqrt{m}$, since $O^u$ is a column stochastic matrix. Therefore,

$$
\begin{aligned}
&\|(\hat{U}^u)^\top O^u - (\hat{\Theta}^u \Pi^u)\| \\
&\leq\quad \|(\hat{U}^u)^\top O^u - (\hat{\Theta}^u \Pi^u)\|_F \\
&\leq\quad 2c_2 \frac{m}{(\pi_{\min}^u)^2} \frac{\epsilon(N, \delta)}{\sigma_{m^d}(P_{1,3}^{H,H})^3 \sigma_m(O^u)^3}
\end{aligned} \tag{15}
$$

We conclude the proof by applying union bound over all $u \in V$. $\qquad\square$

## F  Putting Everything Together – Proof of Theorem 2

*Proof.* (Of Theorem 2) (1) We first give the recovery accuracy of observation matrices. The final step of recovery is $\hat{O}^u = \hat{U}^u \hat{\Theta}^u$. Note that if $N$ is at least $C \max(\frac{D^2}{\sigma_2^2 \sigma_3^2} \ln \frac{D}{\delta}, \frac{m}{\sigma_1^2 \sigma_2^2} \ln \frac{D}{\delta}, \frac{m^2}{\sigma_1^6 \sigma_3^6 \pi_{\min}^3} \ln \frac{D}{\delta})$, then Assumption 3 holds, hence conditioned on event $E$, we have

$$
\begin{aligned}
&\|\hat{U}^u (\hat{U}^u)^\top O^u - O^u\| \\
&=\quad \|\hat{U}^u (\hat{U}^u)^\top O^u - \hat{U}^u (U^u)^\top O^u\| \\
&\leq\quad \|\hat{U}^u (\hat{U}^u)^\top - \hat{U}^u (U^u)^\top\| \|O^u\| \\
&\leq\quad \frac{2\sqrt{m} \epsilon(N, \delta)}{\sigma_m(P_{1,2}^{u,u})}
\end{aligned} \tag{16}
$$

where the first inequality is by the fact that $\|A \cdot B\| \leq \|A\| \|B\|$, the second inequality follows from the fact that $\|O^u\| \leq \sqrt{m}$ and item (1) of Lemma 7.

Meanwhile, by Corollary 2, we have

$$
\begin{aligned}
&\|\hat{U}^u (\hat{U}^u)^\top O^u - \hat{U}^u \hat{\Theta}^u \Pi^u\| \\
&\leq\quad \|(\hat{U}^u)^\top O^u - \hat{\Theta}^u \Pi^u\| \\
&\leq\quad 2c_2 \frac{m}{(\pi_{\min}^u)^2} \frac{\epsilon(N, \delta)}{\sigma_{m^d}(P_{1,3}^{H,H})^3 \sigma_m(O^u)^3}
\end{aligned}
$$

The above two facts let us conclude that provided the size of sample $N$ is at least $C \max(\frac{m}{\sigma_2^2 \sigma_1^8 \epsilon^2} \ln \frac{D}{\delta}, \frac{m^2}{\sigma_3^6 \sigma_1^{14} \pi_{\min}^4 \epsilon^2} \ln \frac{D}{\delta})$ (where we choose $C$ large enough),

$$
\begin{aligned}
&\|O^u - \hat{O}^u \Pi^u\| \\
&\leq\quad \|\hat{U}^u (\hat{U}^u)^\top O^u - O^u\| + \|\hat{U}^u (\hat{U}^u)^\top O^u - \hat{U}^u \hat{\Theta}^u \Pi^u\| \\
&\leq\quad \frac{2\sqrt{m} \epsilon(N, \delta)}{\sigma_m(P_{1,2}^{u,u})} + 2c_2 \frac{m}{(\pi_{\min}^u)^2} \frac{\epsilon(N, \delta)}{\sigma_{m^d}(P_{1,3}^{H,H})^3 \sigma_m(O^u)^3} \\
&\leq\quad \min_{v \in V} \sigma_m(O^v)^4 \epsilon/32 \\
&\leq\quad \epsilon
\end{aligned} \tag{17}
$$

where the first inequality is by triangle inequality, the second inequality is by Equations (15) and (16), the third inequality follows from the choice of $N$, in the last inequality we use the fact that $\sigma_m(O^u) \leq 1$. Therefore by Equation (17) and Theorem 3,

$$\sigma_m(\hat{O}^u \Pi^u) \geq \sigma_m(O^u) - \min_{v \in V} \sigma_m(O^v)^4 \epsilon/32 \geq \sigma_m(O^u)/2 \tag{18}$$

(2) We now provide guarantees on the accuracy of transition probabilities and initial probabilities. In particular, we prove $\|\hat{Q}^u - Q^u(\Pi^u, \Pi^u, \Pi^{\pi(u)})\| \leq \epsilon$, the other three inequalities can be handled similarly. As we have already seen from Equation (17), for all $u \in V$,

$$\|(O^u)^{T\dagger} - (\hat{O}^u \Pi^u)^{T\dagger}\|$$
$$\leq 2\max(\|(\hat{O}^u)^\dagger\|^2, \|(\Pi^{uT}(O^u))^\dagger\|^2)\|O^u - \hat{O}^u \Pi^u\|$$
$$\leq \min_{v \in V} \sigma_m(O^v)^2 \epsilon/16$$

where the first inequality is by Theorem 5, the second inequality uses the fact that $\|(\hat{O}^u)^\dagger\| = 1/\sigma_m(O^u)$, $\|(\hat{O}^u \Pi^u)^\dagger\| = 1/\sigma_m(\hat{O}^u \Pi^u)$ and Equation (18).

Conditioned on event $E$, by the choice of $N$, it is also true that the cooccurence tensor $\hat{P}_{2,2,1}^{u,\pi(u),u}$ is such that

$$\|\hat{P}_{2,2,1}^{u,\pi(u),u} - P_{2,2,1}^{u,\pi(u),u}\| \leq \min_{v \in V} \sigma_m(O^v)^3 \epsilon/32 \tag{19}$$

Therefore,

$$\|Q^u - \hat{Q}^u(\Pi^u, \Pi^{\pi(u)}, \Pi^u)\|$$
$$= \|P_{2,2,1}^{u,\pi(u),u}((O^u)^{T\dagger}, (O^{\pi(u)})^{T\dagger}, (O^u)^{T\dagger}) - \hat{P}_{2,2,1}^{u,\pi(u),u}((\hat{O}^u \Pi^u)^{T\dagger}, (\hat{O}^{\pi(u)} \Pi^{\pi(u)})^{T\dagger}, (\hat{O}^u \Pi^u)^{T\dagger})\|$$
$$\leq \|(P_{2,2,1}^{u,\pi(u),u} - \hat{P}_{2,2,1}^{u,\pi(u),u})((O^u \Pi^u)^{T\dagger}, (O^{\pi(u)} \Pi^{\pi(u)})^{T\dagger}, (O^u \Pi^u)^{T\dagger})\|$$
$$+ \|\hat{P}_{2,2,1}^{u,\pi(u),u}((O^u)^{T\dagger} - (\hat{O}^u \Pi^u)^{T\dagger}, (O^{\pi(u)})^{T\dagger}, (O^u)^{T\dagger})\|$$
$$+ \|\hat{P}_{2,2,1}^{u,\pi(u),u}((\hat{O}^u \Pi^u)^{T\dagger}, (O^{\pi(u)})^{T\dagger} - (\hat{O}^{\pi(u)} \Pi^{\pi(u)})^{T\dagger}, (\hat{O}^u)^{T\dagger})\|$$
$$+ \|\hat{P}_{2,2,1}^{u,\pi(u),u}((\hat{O}^u \Pi^u)^{T\dagger}, (\hat{O}^{\pi(u)} \Pi^{\pi(u)})^{T\dagger}, (O^u)^{T\dagger} - (\hat{O}^u \Pi^u)^{T\dagger})\|$$
$$\leq \|(P_{2,2,1}^{u,\pi(u),u} - \hat{P}_{2,2,1}^{u,\pi(u),u})\| \max_{v \in V} \|O^{v\dagger}\|^3 + \|\hat{P}_{2,2,1}^{u,\pi(u),u}\| \cdot \|(O^u)^{T\dagger} - (\hat{O}^u \Pi^u)^{T\dagger}\|(\max_{v \in V} \|O^{v\dagger}\|^2 +$$
$$\max_{v \in V} \|O^{v\dagger}\| \max_{v \in V} \|\hat{O}^{v\dagger}\| + \max_{v \in V} \|\hat{O}^{v\dagger}\|^2)$$
$$\leq \epsilon$$

where the first inequality is by triangle inequality, the second inequality is by the fact that $\|T(A, B, C)\| \leq \|T\|\|A\|\|B\|\|C\|$, the third inequality is by Equations (19) and (17).

$\square$

## G  Matrix Perturbation Lemmas

**Theorem 3** (Weyl's Theorem). *If $A$, $E$ are matrices in $\mathbb{R}^{m \times n}$ with $m \geq n$. Then,*
$$|\sigma_i(A + E) - \sigma_i(A)| \leq \|E\|$$

**Theorem 4** (Wedin's Theorem). *If $A$, $E$ are matrices in $\mathbb{R}^{m \times n}$ with $m \geq n$. Let $A$ have singular value decomposition:*

$$\begin{pmatrix} U_1^\top \\ U_2^\top \\ U_3^\top \end{pmatrix} A \begin{pmatrix} V_1 & V_2 \end{pmatrix} = \begin{pmatrix} \Sigma_1 & 0 \\ 0 & \Sigma_2 \\ 0 & 0 \end{pmatrix}$$

*Let $\tilde{A} = A + E$ have the singular value decomposition:*

$$\begin{pmatrix} \tilde{U}_1^\top \\ \tilde{U}_2^\top \\ \tilde{U}_3^\top \end{pmatrix} \tilde{A} \begin{pmatrix} \tilde{V}_1 & \tilde{V}_2 \end{pmatrix} = \begin{pmatrix} \tilde{\Sigma}_1 & 0 \\ 0 & \tilde{\Sigma}_2 \\ 0 & 0 \end{pmatrix}$$

*If there is $\delta > 0$, $\alpha > 0$ such that $\min_i \sigma_i(\tilde{\Sigma}_1) \geq \alpha + \delta$, $\max_i \sigma_i(\Sigma_2) \leq \alpha$, then*

$$\|\sin \Phi\| \leq \frac{\|E\|}{\delta}$$

*where $\Phi$ is the matrix of principal angles between $\mathrm{range}(U_1)$ and $\mathrm{range}(\tilde{U}_1)$.*

**Theorem 5.** *If $A$, $E$ are matrices in $\mathbb{R}^{m \times n}$ with $m \geq n$, let $\tilde{A} = A + E$. Then,*

$$\|\tilde{A}^\dagger - A^\dagger\| \leq 2\max(\|\tilde{A}^\dagger\|^2, \|A^\dagger\|^2)\|E\|$$

## H  Compressed observation matrices produced by Spectacle-Tree for eight ENCODE cell types

Figure 3: The compressed observation matrices estimated by Spectacle-Tree for all eight ENCODE cell types studied, other than GM12878 which is presented in the main manuscript.