[Reviews · NeurIPS 2015]

Submitted by Assigned_Reviewer_1

This paper proposes to solve the parameter learning problem in tree structured HMM using spectral method. Taking advantage of the tree structure, the authors reduced complexity from x^D to x^d where D is the number of nodes in the tree and d is the depth. This represents a significant saving in the case of d << D (for example, for balanced trees we have d = O(log(D)).

The paper is well written, where both the goal and the approach is well motivated and the model is clearly explained as well. The authors summarized their contribution in two techniques called the skeletensor and the product projection, both of which are new to me.

The experimental results seem strong, although I'm not familiar with the settings and the dataset. One thing that is objectionable is that SMF seems too weak as far as a baseline is concerned. Perhaps regular EM makes a good baseline, since it seems to me that EM can also use the tree structure instead of working in an exponentially large state space by expanding the tree. So I'm not convinced that EM should be dismissed as very slow without providing more arguments. I understand that space is constrained, but experiments on synthetic data, with generic evaluations (like parameter distance, loglikelihod) would help people in the ML community not familiar with the biology task better evaluate this method. Overall, while I'm not entirely sure how important tree HMM for the application, this paper made clear, original progress on an interesting problem, and the techniques developed can probably be used in other settings. This paper should be accepted.
Summary: A well written paper that made a well-defined contribution of speeding up learning of tree structured HMM.

I'm not totally confident about the importance of this problem and am slightly bothered by the absence of EM in the experimental results.

Submitted by Assigned_Reviewer_2

The tree-structured hidden states are very interesting, while the extension from the regular HMM spectral learning seems rather minor and incremental. More importantly, the motivation and problem setting are not well explained. I am not sure why HMM is needed for the data on chromatin marks. As implied by the applications of HMM, such as speech recognition, thus far, each instance of the HMM data has a variable length, while the given data is a simple binary matrix (fixed length). Also I cannot see why the tree dependency is required for the data. I think these points should be clearly mentioned, particularly in Introduction.
Summary: See below.

Submitted by Assigned_Reviewer_3

The proposed method seems original and helps to solve an important current problem in comparative epigenomics. The ability to handle the background state well in comparison with the previsou method looks interesting. The model is validated on real datasets from ENCODE project, but more experiments results seem necessary to fully validate the model.

- Simulation study is lacking. The accuracy of the inference is shown only in terms of promoter site prediction

- Comparison with other plain HMM models (e.g. ChromHMM) would also be interesting as a baseline, to highlight the benefit of emposing tree structure in hidden states.

Summary: This paper presents a latent variable model and an efficient spectral algorithm with tree-structured HMM for comparative chromatin state segmentation problem. The idea seems novel and effective, but thourough experimental validation is lacking.

Submitted by Assigned_Reviewer_4

The paper apply the method called 'spectral method', which can be seen as a method of moments technique. The main interest of such method is its statistical consistency.

The framework is a THS-HMM (tree-structure hidden states HMM), an object which is used to study chromatin data in multiple cells.

The authors explain how to avoid dealing with an exponential number of states, what would occur if the method of moments would be applied naively. They present three techniques that would allow significant computing time reduction.

The authors also provide statistical consistency results.

Some experiments are provided, which show that the technique presented in the paper allows improvements on some real biological tasks.

The paper seems to be a solid contribution, with a strong theoretical part, an an experimental which seems solid also, though I'm not a specialist of this domain.

-quality: good quality, theoretical and experimental

-clarity: the paper is quite clear, but the main justification of this method (the cost of applying the standard method naively) is not really discussed, and may remain unclear to reader that are not familiar with those techniques.

-originality: average, as most of the methods used are known. I would say that this paper is mainly incremental, as I'm not convinced by the follow-up argument.
Summary: Good paper, though mainly incremental

Author Feedback
Author rebuttal: We thank all reviewers for their feedback.

Reviewer 1:

Regarding the importance of the problem, HMMs are a standard model for biological sequences (including proteins, DNA etc, not just chromatin) and trees are a standard model for relationships between cell types, species, individuals etc. Thus the model extends beyond the specific application in chromatin to a large number of important biological problems. We note that Reviewer 2 wrote that it is "an important current problem in comparative epigenomics" and Reviewer 6 wrote that it is "an important practical problem".

Regarding the lack of comparison to EM, Biesinger et al. (BMC Bioinformatics 2013) performed extensive comparisons of structured mean field (SMF) variational inference to EM, as well as mean field variational inference and loopy belief propagation, for tree-structured HMMs. They showed convincingly that SMF outperforms the other approximations on both artificial and real data and concluded that "The closeness between SMF free energy and the exact log likelihood indicates that the SMF method captures the majority of correlation between variables." Thus we believe it is appropriate to use SMF as the baseline instead of EM in our experiments. Furthermore, Song and Chen (Genome Biology 2015) performed extensive comparisons of spectral learning with EM (e.g. as implemented in the software ChromHMM) for regular HMMs and concluded that spectral learning outperforms EM for real biological data. Thus we also omit the comparison to EM for regular HMMs.

Regarding simulated data, a recent manuscript by Mattila, Rojas and Wahlberg (arXiv 1507.06346) evaluated spectral learning for HMMs on simulated data and concluded that its performance depends on the specific data set but spectral learning is often preferable for large number of observations. In particular, this suggests that spectral learning should be a good method for real biological data.

Reviewer 2:

Please see our response to Reviewer 1 regarding simulations and the extensive comparisons to regular HMMs (including the ChromHMM software) in Song and Chen (Genome Biology 2015).

Reviewer 3:

Regarding the cost of the standard spectral method, we apologize that our manuscript was not clear. In theory, the cost of applying the standard method naively is discussed in lines 188-191, which is Omega(n^D m^D). The reason is that the meta observation matrix to be recovered (using naive method) is an explicit n^D * m^D matrix: the meta state at time t is (z_t^v)_{v \in V}, which takes m^D values, and meta observation at time t is (x_t^v)_{v \in V}, which takes n^D values. In the experimental section we wrote that the standard spectral method could not run on our workstation because it ran out of memory (despite optimized use of sparse linear algebra packages). We will clarify this point in the final version.

Regarding the novelty of the methods, we respectfully note that our Skeletensor method is quite different from the symmetrization procedure in the standard spectral algorithm for HMMs based on tensor decomposition. In particular handling the asymmetry of the matrices is considerably more delicate in our algorithm. We would also like to point out that Reviewer 1 wrote that both the Skeletensor and Product Projections arguments were novel to him/her.

Reviewer 4:

We would like to point out that the model we study is applicable to many problems in biology as described in our response to Reviewer 1.

Reviewer 5:

Regarding the novelty of the methods, please see our response to Reviewer 3.

Thank you for the suggestions - we will improve the motivation of the model in the final version. Briefly, biological sequences are not fixed length - for example, different species like mouse and pig have different lengths from human.

Also, trees are a classic probabilistic model in biology and are commonly used to model relationships between cell types or species. In addition to Biesinger et al. BMC Bioinformatics 2013 cited above, other prominent examples that combine trees and (generalizations of) HMMs in biology include Siepel et al. Evolutionarily conserved elements in vertebrate, insect, worm, and yeast genomes, Genome Research, 2005 and Pedersen et al. Identification and classification of conserved RNA secondary structures in the human genome, PLoS Computational Biology, 2006. Previous methods in biology typically used the EM algorithm.